# Hippocampal ensembles represent sequential relationships among an extended sequence of nonspatial events

Babak Shahbaba [1,2,3], Lingge Li[2,3], Forest Agostinelli [2,3,4], Mansi Saraf [5,6], Keiland W. Cooper [5,6], Derenik Haghverdian [2], Gabriel A. Elias [5,6], Pierre Baldi [2,3] & Norbert J. Fortin [1,5,6✉]

The hippocampus is critical to the temporal organization of our experiences. Although this fundamental capacity is conserved across modalities and species, its underlying neuronal mechanisms remain unclear. Here we recorded hippocampal activity as rats remembered an extended sequence of nonspatial events unfolding over several seconds, as in daily life episodes in humans. We then developed statistical machine learning methods to analyze the ensemble activity and discovered forms of sequential organization and coding important for order memory judgments. Specifically, we found that hippocampal ensembles provide significant temporal coding throughout nonspatial event sequences, differentiate distinct types of task-critical information sequentially within events, and exhibit theta-associated reactivation of the sequential relationships among events. We also demonstrate that nonspatial event representations are sequentially organized within individual theta cycles and precess across successive cycles. These findings suggest a fundamental function of the hippocampal network is to encode, preserve, and predict the sequential order of experiences.

---

[1] Data Science Institute, University of California, Irvine, CA, USA. [2] Department of Statistics, University of California, Irvine, CA, USA. [3] Department of Computer Science, University of California, Irvine, CA, USA. [4] Artificial Intelligence Institute, University of South Carolina, Columbia, SC, USA. [5] Center for the Neurobiology of Learning and Memory, University of California, Irvine, CA, USA. [6] Department of Neurobiology and Behavior, University of California, Irvine, CA, USA. ✉email: norbert.fortin@uci.edu

In humans, the hippocampus is known to play a key role in the temporal organization of memory and behavior. This includes our ability to remember when past experiences occurred[1–4], but also extends to our ability to use information about past experiences to imagine or predict future outcomes[5,6]. Considerable research indicates that this capacity is conserved across species and applies across spatial and nonspatial modalities[3,4,6], yet the neural mechanisms supporting it remain poorly understood. The emerging conceptual framework suggests that the propensity of the hippocampal network to generate and preserve sequential patterns of activity may underlie this fundamental capacity[4,6–10], a view supported by two main lines of electrophysiological evidence. First, hippocampal ensemble activity tends to exhibit sequential firing fields during the presentation of nonspatial stimuli or inter-stimulus intervals (also known as "time cell" activity)[11–13], which has been shown to provide a strong temporal signal within such task events[14,15]. Second, hippocampal neurons have been shown to code for sequences of spatial locations under different experimental conditions[6,16–18]. Of particular interest here is evidence that hippocampal activity can represent sequences of previous, current, and upcoming locations when animals run on a maze[19–23] or pause at a decision point (vicarious trial-and-errors)[24], conditions in which the hippocampal network displays prominent theta oscillations and is thought to be engaged in online processing of upcoming decisions and goals[6,9]. However, the key piece of evidence directly linking this sequence coding framework with our fundamental ability to remember and predict event sequences across spatial and nonspatial modalities remains missing. Specifically, it is critical to demonstrate that these coding properties: (i) extend to sequences of nonspatial events unfolding over several seconds, as in daily life episodes, and (ii) are linked to the successful retrieval of such event sequences.

To address this important issue, we recorded hippocampal ensemble activity as rats performed a challenging nonspatial sequence memory task with established parallels in humans[25]. Taking advantage of this unique behavioral approach, we began by examining how the organization of sequential firing fields varied throughout this extended sequence of discontiguous events (series of odor stimuli). We then developed statistical machine learning methods, including deep learning approaches, to reveal the sequential structure in which the representations of different types of information varied within individual stimulus presentations (i.e., within 1.2 s). We present compelling evidence of nonspatial forms of sequential organization and coding in hippocampal ensembles linked with correct sequence memory judgments. First, we found that hippocampal ensembles provided significant temporal information during individual stimulus presentations, which was primarily stimulus-specific but also reflected sequential relationships among stimuli through a temporal lag effect, and that this temporal coding extended across the full sequence of stimuli unfolding over several seconds. Second, using a latent representation learning approach, we also found that the ensemble activity simultaneously and sequentially differentiated distinct types of trial-specific information within stimulus presentations, including the stimulus presented, its temporal order, and whether the animal correctly identified the trial type. Third, using a neural decoding approach to quantify the decoding probability of each stimulus in the sequence, we discovered that the sequential relationships among these nonspatial events (separated by several seconds in real time) were reactivated during individual stimulus presentations (within 1 s), providing direct evidence that theta-associated forward reactivation extends beyond the domain of spatial information. Finally, using a simpler decoding model with a higher temporal resolution, we found that these sequential relationships can even be compressed within a single theta cycle, that the information represented by individual neurons precessed across cycles, and confirmed the sequential reactivation pattern within trials observed with the previous model. Collectively, these results suggest a fundamental function of the hippocampus is to simultaneously represent the sequential order of experience not only in real-time, including internal representations of the temporal context of events and of different forms of task-critical information, but also at a higher level of abstraction, including temporally compressed representations extracting task-critical sequential relationships among events separated in time by several seconds. These findings are consistent with, and provide potential neuronal mechanisms for, the critical role of the hippocampus in temporally organizing our past experiences and future behavior.

## Results

We trained rats to perform a nonspatial sequence memory task, which shows strong behavioral correspondence in rats and humans[25]. In rats, this task involves repeated presentations of sequences of odors in a common odor port (e.g., odors ABCDE) and requires subjects to determine whether each odor is presented "in sequence" (InSeq; e.g., AB<u>C</u>…) or "out of sequence" (OutSeq; e.g., AB<u>D</u>…) to receive a water reward (Fig. 1a). Importantly, this InSeq/OutSeq judgment is performed on a trial-by-trial basis (each trial corresponds to the presentation of one odor in the sequence of five), such that each correctly identified odor was followed by a reward and an incorrect response resulted in early termination of the sequence. After reaching criterion on the task, animals were implanted with a microdrive and, over a few weeks, tetrodes were gradually lowered into the pyramidal cell layer of the dorsal CA1 region of the hippocampus. Neural activity (spikes and local field potentials; LFP) was then recorded as animals performed the task (Fig. 1b; see ref. [26] for a detailed analysis of single-cell and LFP activity from this dataset).

Here we focus on identifying the temporal dynamics of CA1 ensemble activity within odor presentations, during which the animals' location is constant and the LFP activity displays prominent theta oscillations[26], that support the memory for sequences of events. Note that each analysis focused on comparisons in which the animal's behavior was consistent across levels, either by only including correctly identified InSeq trials or by focusing on specific time windows in which the behavior was matched across conditions. To balance statistical power across analyses, we concentrated on the first four stimuli in the sequence to account for the decrease in sampling with sequence position produced by incorrect responses (trial and neuron counts are provided in Supplementary Tables 1 and 2). We also took a rigorous and conservative approach to data inclusion, sampling, and pooling to maximize the reproducibility of the findings (see the corresponding section in "Methods").

**CA1 ensemble activity provides a temporal signal that carries event-specific information and bridges across event sequences.** Consistent with previous reports[11–15], hippocampal activity showed sequences of firing fields ("time cell" activity) during stimulus presentations (Fig. 2a) as well as during intervals between stimulus presentations (Supplementary Fig. 1). Here we extend these findings by determining how this form of temporal coding interacts with stimulus identity and sequence position (lag), and whether it extends across sequences of stimuli. Note that we visualized this sequential organization using peri-stimulus time histograms (PSTHs) sorted by peak firing latency (see "Methods" section), but quantified its degree of temporal coding by implementing a Bayesian model to reconstruct time using the

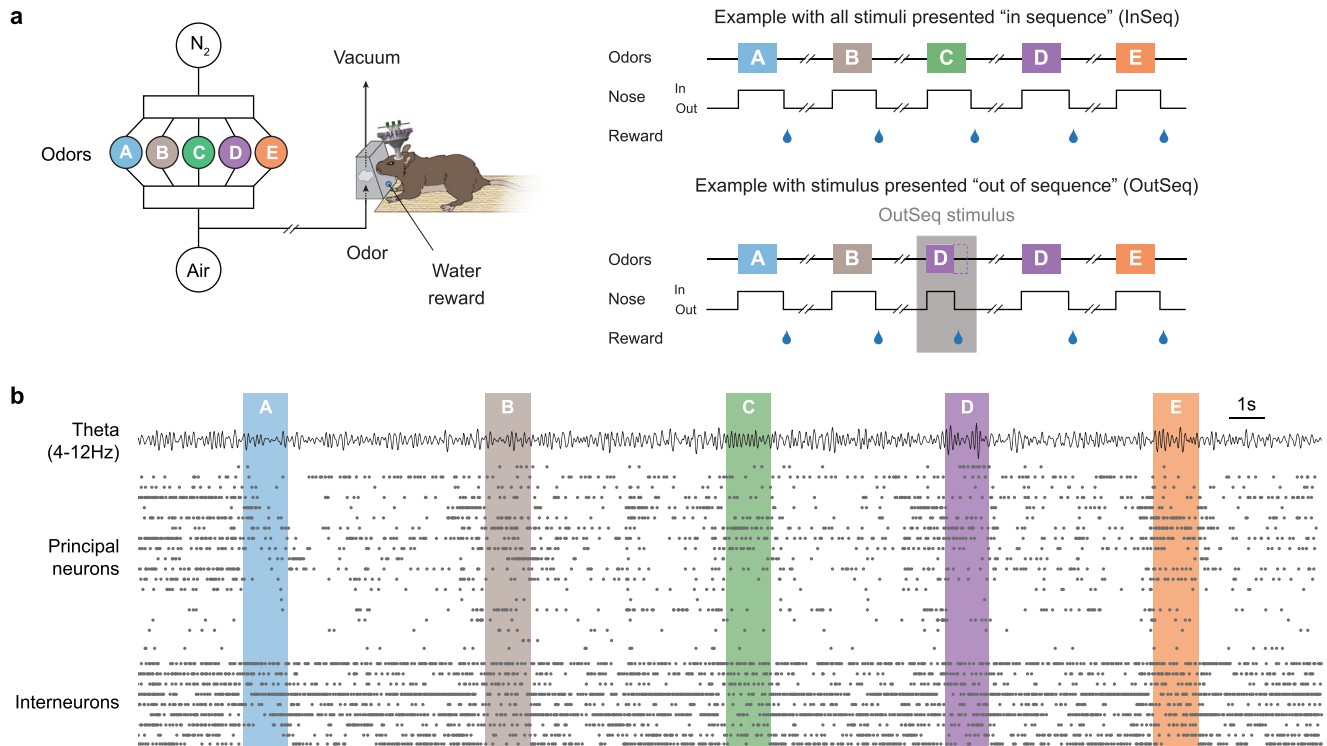

**Fig. 1 Neural activity was recorded from hippocampal region CA1 as animals performed a complex nonspatial sequence memory task. a** The task involves repeated presentations of sequences of nonspatial events (odor stimuli) and requires subjects (rats) to determine whether each stimulus is presented "in sequence" (InSeq; e.g., ABC...) or "out of sequence" (OutSeq; e.g., ABD...). Using an automated odor delivery system (left), self-paced sequences of five odors (odor A = sky blue, B = brown, C = green, D = purple, E = orange) were presented in the same odor port (median interval between consecutive odors ~5 s). In each session, the same sequence was presented multiple times (right), with approximately half the presentations including all InSeq trials (top) and the other half including one OutSeq trial (bottom). Each odor presentation was initiated by a nosepoke and rats were required to correctly identify each odor as either InSeq (by holding their nosepoke response until a tone signaled the end of the odor at 1.2 s) or OutSeq (by withdrawing their nose before the signal; <1.2 s) to receive a water reward. Incorrect InSeq/OutSeq judgments resulted in termination of the sequence. **b** Example ensemble activity (putative principal neurons and interneurons) and theta oscillations (bandpass: 4–12 Hz) from representative subject during one sequence presentation. Principal neurons and interneurons were separately sorted by their peak firing time in relation to the port entry triggering delivery of odor A.

ensemble activity alone (the model is agnostic to the PSTH sorting and uses a cross-validation approach; see "Temporal coding analyses" and "Control analyses" in "Methods" section).

First, we found that these sequential patterns of activity during odor presentations were stimulus-specific and linked to correct order memory judgments. For each odor type, a strong temporal organization was observed when neurons were sorted by their peak firing latency relative to the port entry triggering that odor, whereas it was significantly reduced when the same sorting of neurons was examined during other odors (Fig. 2a). We started by examining this temporal organization in a model-free way by comparing the similarity of PSTHs. To do so, we correlated the PSTHs from individual trials (the neuron by time bin matrix from each odor presentation; see "Methods" section) and compared the correlation coefficients using ANOVAs. We found that PSTHs were highly correlated across trials of the same odor type (PSTHs along the diagonal in Fig. 2a; $r_{OdorA} = 0.7797$, $r_{OdorB} = 0.7773$, $r_{OdorC} = 0.7632$, and $r_{OdorD} = 0.7631$), and were significantly less correlated to trials of another odor type (other PSTHs in the same row, which keep the neuron sorting constant; all one-way ANOVAs and Dunnett's posthoc tests $p$ values < 0.0001; Supplementary Fig. 2a). We then performed the same comparisons using the Bayesian model, to directly quantify temporal coding, and observed the same pattern. In fact, our ability to accurately decode time based on the ensemble activity (accuracy of reconstructed time) was significantly higher when model training

and decoding was performed on trials of the same type (e.g., train and decode using different subsets of odor A trials) than across trial types (e.g., train on odor A trials and decode B, C, or D trials; all one-way ANOVAs and Dunnett's posthoc tests $p$ values < 0.0001; Supplementary Fig. 2b). Importantly, reconstructed time accuracy was significantly higher before correct responses than incorrect responses (Kolmogorov–Smirnov $D = 0.3232$, $p = 0.0149$), suggesting this form of activity supports order memory decisions. Note that this effect was tested on OutSeq trials for adequate sampling of correct and incorrect responses, using the 250 ms window before port entry to avoid the potential influence of the OutSeq stimulus, and that the result held when the comparison was downsampled (Kolmogorov–Smirnov $D = 0.364$, $p = 0.013$; see "Methods" section). Finally, it should also be noted that stimulus-specificity was visibly weaker in the first 250 ms period, suggesting a subset of neurons reflected a shared experience across trials (e.g., sampling the air in the port until the odor is identified; see next section).

Second, we found that this temporal coding systematically varied across (non-preferred) stimuli. In fact, consistent with the temporal context model[27,28], reconstructed time accuracy varied by lag ($F_{(6,873)} = 166.9$, $p < 0.0001$; Fig. 2c). It was highest for lags of 0 (i.e., train and decode using different subsets of trials from same odor type; comparisons with lag 0: all Dunnett's posthoc tests $p$ values < 0.0001) and showed a significant linear decline across lags of 1, 2, and 3 in the positive direction (e.g., train on B,

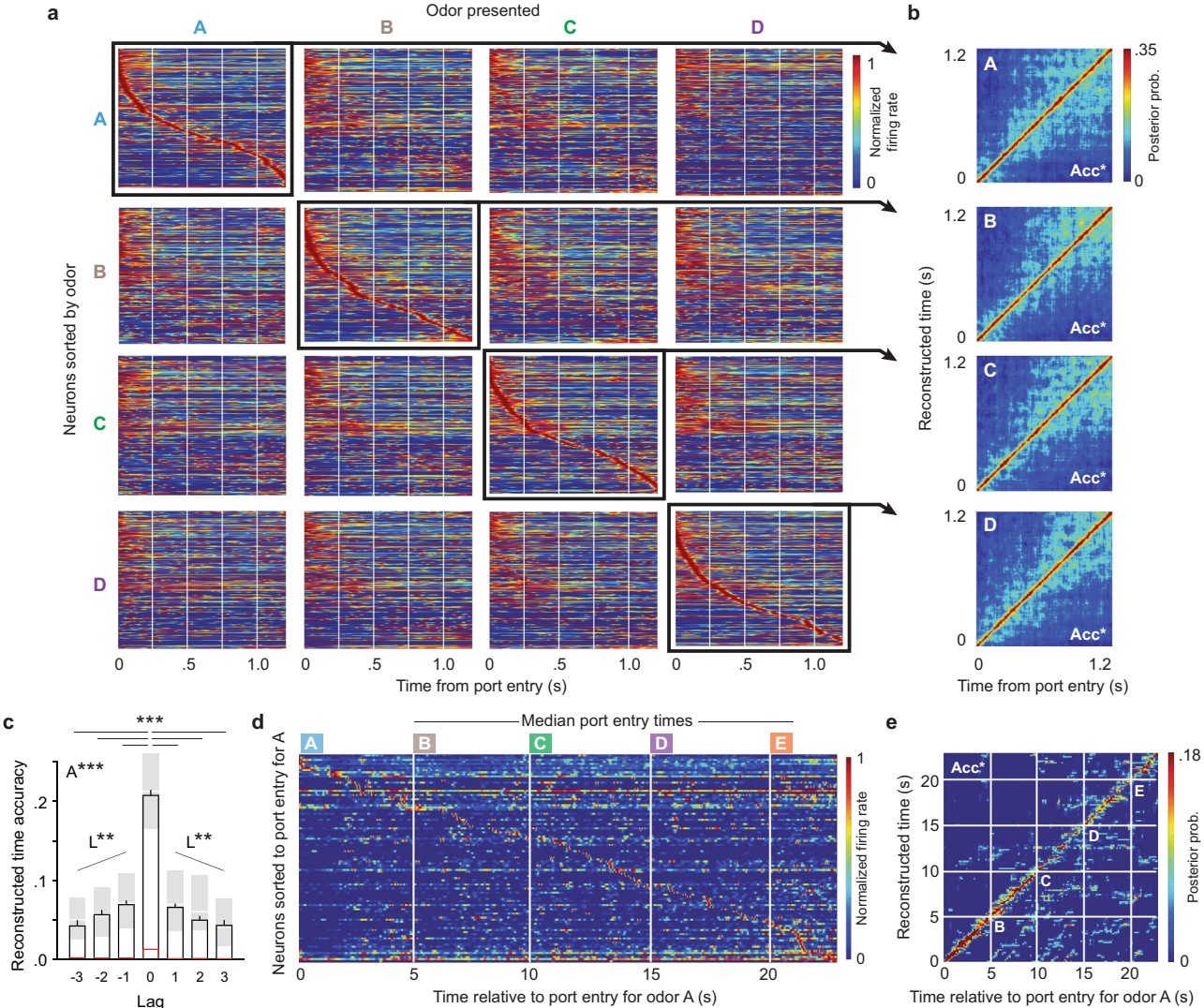

**Fig. 2 CA1 ensembles display sequences of firing fields ("time fields") within individual events, which vary by the associated stimulus and sequence position, and across event sequences. a** Stimulus-specificity in sequential firing fields. Peri-stimulus time histograms (PSTH; 150 ms gaussian) show the normalized firing rate of all active neurons for each odor type (correct InSeq trials only; active neurons for each odor: A = 267, B = 192, C = 209, D = 223; neurons collapsed across five subjects, one session per subject). To visualize how sequential firing fields varied across odor types, PSTHs are shown for each odor presented (in columns), with neurons sorted by their time of peak firing relative to the port entry for each odor (in rows). **b** Temporal coding accuracy for each odor type was above-chance levels. Plots show reconstructed time estimates obtained from each animal's PSTH (correctly sorted for each odor; i.e., diagonal of panel **a** but separated by animal) and averaged across subjects. **c** Accuracy of reconstructed time varied by the lag between the odor type used to train the model and the odor type in which time was reconstructed (e.g., when the model was trained on odor B trials, decoding time during odor B, C, D or E trials represented lags of 0, 1, 2, or 3, respectively). Black bars depict mean ± SEM (Lag 0: n = 220; Lag ± 1: n = 165; Lag ± 2: n = 110; Lag ± 3: n = 55; trial data pooled across subjects), shaded regions the Q1–Q3 range (median denoted by white line), and red lines the permuted chance levels. Lag data show a significant one-way ANOVA ($F_{(6,873)} = 166.9$, $p < 0.0001$; A***), difference between lag 0 and all other lags (two-tailed Dunnett's posthoc tests, adjusted for multiple comparisons; ***$p < 0.0001$), and linear trends (positive lag: $F_{(1,327)} = 12.93$, $p = 0.0004$; negative lag: $F_{(1,327)} = 12.20$, $p = 0.0005$; L**). **d** The sequential organization of firing fields extended across the full sequence of odors. Since the task is self-paced, the PSTH (250 ms gaussian) shows data from a single subject with median times for odors B, C, and D across sequences. **e** Accuracy of reconstructed time across the sequence of odors (PSTH from panel **d**) was above-chance levels. Acc*, reconstructed time accuracy significantly above-chance levels (determined by random permutations). Color coding of odor types: odor A = sky blue, B = brown, C = green, D = purple, E = orange.

decode C, D, or E; $F_{(1,327)} = 12.93$, $p = 0.0004$) and negative direction (e.g., train on D, decode C, B, or A; $F_{(1,327)} = 12.20$, $p = 0.0005$). It is also important to note that reconstructed time accuracy was significantly above-chance levels across all lags (Fig. 2c; all one-sample $t$-tests $p$ values < 0.05; chance-level determined by permutations). These results suggest that, although predominantly stimulus-specific, this temporal organization is partially shared between stimuli and decreases with the distance from the preferred stimulus.

Finally, to determine if this temporal organization extended beyond individual stimuli or intervals, we examined the ensemble activity across the sequence of odors. Despite the variable time at which odors were presented (the task is self-paced so the time between odor presentations varied across sequences), the sequential organization of firing fields can be observed across the whole sequence (Fig. 2d). Importantly, this activity provides significant information about time within the sequences of events (Fig. 2e), as reconstructed time accuracy was significantly higher

than chance levels for each subject ($t_{(11)}$ = 7.043, $t_{(6)}$ = 4.730, $t_{(13)}$ = 7.941, $t_{(11)}$ = 6.174, $t_{(9)}$ = 7.398; all $p$ values < 0.01; same pattern of results was observed using shorter and longer time bins, or when collapsing across subjects; Supplementary Fig. 3). Collectively, these findings suggest that this form of sequential organization in CA1 ensemble activity can provide a task-critical temporal signal during event presentations, during the intervals separating them, as well as across the full sequences of events.

**The differentiation of stimulus identity, temporal order, and trial outcome information varies within individual events**. Next, we examined the temporal dynamics by which different types of task-critical information were represented within trials (wherein a trial corresponds to one odor presentation). As detailed below, we developed a different model to quantify this because of the need to reduce the dimensionality of the ensemble activity using an unsupervised approach, which is incompatible with the Bayesian model used in the previous section. Because the representations of nonspatial stimuli in hippocampal neurons are weaker and more complex than the representations of spatial locations[29,30], we used deep learning approaches to quantify the information represented in the ensemble activity. We began by identifying the underlying structure of ensemble activity at different moments within trials using a latent representation learning approach in which the model is not provided with any information about trial type. More specifically, we used an autoencoder, which can be viewed as a nonlinear counterpart to principal component analysis, to encode the spiking activity data into a two-dimensional latent space (by using only two neurons at the bottleneck of the neural network architecture; see "Latent representation analyses" in "Methods" section). This approach allowed us to visualize a two-dimensional representation of the ensemble activity at different time windows within trials, to which trial labels were subsequently added for posthoc analysis (Fig. 3a). The resulting cluster plots show the location of the neural activity in the two-dimensional space defined by the activation level of the two nodes in the middle layer (bottleneck) of the autoencoder. The degree to which the clusters mapped onto each type of trial information was quantified using a k-nearest neighbor approach ($k = 2$; with cross-validation) across animals, with high classification accuracy indicating high differentiation of a specific type of information (Fig. 3c). For statistical comparisons, we determined the 95% confidence interval for chance-level classification accuracy for each type of information and time window using permutations. Accuracy values above the confidence intervals were considered statistically significant. Due to the different response dynamics for correct InSeq trials (i.e., hold until the signal) and OutSeq trials (i.e., withdraw before signal), we focused on four epochs common to both trial types. Specifically, as most OutSeq responses occurred after 500 ms (89.5% OutSeq responses were >500 ms; mean ± SD: 751 ms ± 229 ms), we examined the first two 250 ms windows after port entry to highlight dynamics early in the trial period. Additionally, we examined the 250 ms windows immediately before and after the port withdrawal response.

While the model did not include trial-specific information, we identified patterns in the data that strongly differentiated information about stimulus (which odor was presented), temporal order (whether the odor was InSeq or OutSeq), and trial outcome (whether the animal correctly identified the trial or not). More importantly, this approach simultaneously captured the temporal dynamics of each type of information within trials. Weak differentiation was observed in the first time window immediately after port entry (0–250 ms), indicating little trial-specific information was present at that point. Although above-chance odor differentiation was observed in that time period,

which likely reflects predictive coding of upcoming odors (see next section), the differentiation of temporal order or trial outcome information was at chance levels. The second time window (250–500 ms after port entry) showed a marked increase in odor differentiation, accompanied by above-chance differentiation of temporal order and trial outcome information. This sharp increase in odor differentiation 250 ms after port entry suggests this effect is mostly driven by the odor itself, not simply information about sequence position. In the third time window (250 ms period before withdrawal response), temporal order differentiation showed a marked increase, with odor and trial outcome differentiation sustaining their levels. This increase in temporal order differentiation immediately before the animals' response is consistent with the main task requirement (to identify each item as InSeq or OutSeq) and with previous single-cell analyses showing hippocampal neurons firing differentially on InSeq vs OutSeq trials[26]. The fourth time window (250 ms period after withdrawal response) showed reduced odor and temporal order differentiation, but preserved trial outcome differentiation (which peaked in the subsequent window). The trial outcome differentiation, when observed during odor presentations, indicates the presence of a pattern in the ensemble activity that is predictive of whether the animal will respond correctly or not on that particular trial; this may reflect the anticipation of the associated reward or error signal, or disrupted representations of the predicted stimulus, currently presented stimulus, or InSeq/OutSeq status of the trial. However, the fact that this differentiation peaks after the withdrawal response suggests that it then also incorporates the neural response to feedback signals (water reward on correct trials, buzzer on incorrect trials).

Finally, a deeper look at the temporal order differentiation revealed that, just as with temporal coding (Fig. 2c), a representation of the temporal context influenced the ensemble dynamics observed here. In other words, the relative distance between clusters reflected the ordinal relations among the odors; for instance, the distance between odor A and B clusters (Lag 1) was smaller than between A and C clusters (Lag 2) and between A and D clusters (Lag 3), and so on. More specifically, at all time points during the trial, cluster distances were significantly different across lag (all one-way ANOVA $p$ values < 0.05; see Supplementary Table 3) with the magnitude of the distance scaling linearly with lag (all linear trend analyses $p$ values < 0.05; Fig. 3b; Supplementary Table 3). Importantly, this effect was observed for both InSeq and OutSeq trials, indicating that temporal order differentiation was not simply a match-mismatch signal. Moreover, the lag effect was strongest (steepest slope) in the periods preceding behavioral responses, suggesting temporal context information was maximally influential on the ensemble state at this point. To summarize, our results show that ensemble activity simultaneously differentiated information about stimulus identity, temporal order, and trial outcome within individual stimulus presentations. Not only was the sequential organization of their respective peaks consistent with the expected flow of task-critical information within a trial, this information was also "multiplexed" with a representation of the trial's temporal context.

**The sequential relationships among discontiguous events are reactivated within individual events**. To follow up on the finding of sustained odor differentiation within trials, we then determined the content of the odor information represented. More specifically, we quantified the decoding of each odor representation at different moments within trials. As detailed below, this quantification required the development of a separate model because of the need for a supervised approach with dimensionality

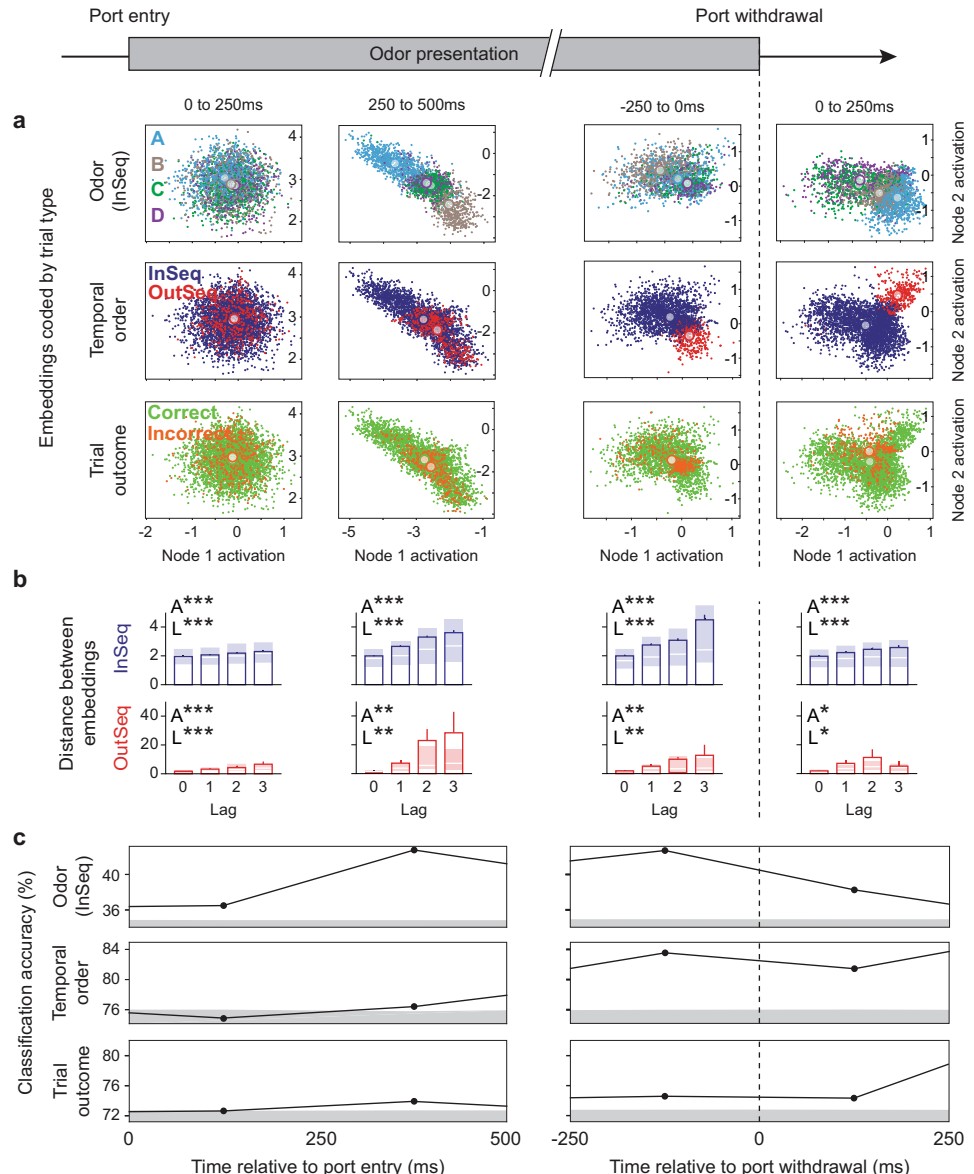

**Fig. 3 Stimulus identity, temporal order, and trial outcome information are simultaneously differentiated, but peak at different times, within events.** A deep learning latent representation approach was used to identify the underlying structure of ensemble activity in four 250 ms windows matching behavior across InSeq and OutSeq trials, to which trial labels were subsequently applied. **a** Differentiation of stimulus (odors: A = sky blue, B = brown, C = green, D = purple; correct InSeq trials only), temporal order (InSeq in navy blue, OutSeq in red; correct trials only), and trial outcome (correct trials in light green, incorrect trials in dark orange) information in example subject. The model reduced the dimensionality of each animal's ensemble activity to two dimensions, which correspond to the activation level of the two nodes in the middle layer of the autoencoder (see "Methods" section). Each point representing a 100 ms slice of spike activity data projected onto this two-dimensional space (white circles indicate cluster centroids). To better visualize cluster separation at different moments within trials, the model was run separately on each 250 ms window (shown in columns; plots are on same scale within a column but are re-scaled across columns). **b** The distance between clusters scaled as a function of lag (the distance between odors in the sequence) for both InSeq and OutSeq trials. Bars depict mean ± SEM (correct trials only; trial data pooled across subjects) and shaded regions the Q1–Q3 range (median denoted by white line). Data from InSeq trials is shown in navy blue (lag 0: $n = 664$; lag 1: $n = 995$; lag 2: $n = 664$; lag 3: $n = 333$) and OutSeq trials in red (lag 0: $n = 46$; lag 1: $n = 32$; lag 2: $n = 26$; lag 3: $n = 14$). Lag data show significant one-way ANOVAs (A*, $p < 0.05$; A**, $p < 0.005$; A***, $p < 0.0001$) and linear trends (L*, $p < 0.05$; L**, $p < 0.005$; L***, $p < 0.0001$; see Supplementary Table 3). **c** Differentiation (mean classification accuracy; trials pooled across subjects) for each type of information relative to chance levels. Since two-dimensional embeddings were specific to each rat's neuronal ensemble, classification accuracy for each trial was calculated using a k-nearest neighbor approach ($k = 2$) to allow trial data to be pooled across subjects. Gray bands represent chance-level classification accuracy (values below +95% CI; determined by random permutations). Vertical dotted line indicates port withdrawal time.

reduction, a joint requirement incompatible with the previous two models. We used a convolutional neural network (CNN) model with a tetrode-wise filtering architecture, which takes both spike data and LFP signals as input (unlike the previous models which were not designed to incorporate LFP), and odor labels as its output (see "Neural decoding analyses" in "Methods" section). During training, the model was supplied trial-identified neural activity from the 150–400 ms window and used multiple non-linear hidden layers to create a map between input and output. After training, the hidden layer that feeds into the output layer

provided a supervised latent representation of the data, which was used to decode odor representations across time windows. For statistical comparisons, we determined the 95% confidence intervals of the decoding probabilities in each time window (which follow a multinomial distribution). To be conservative, decoding probabilities with non-overlapping confidence intervals were considered significantly different.

Our results show that the representations of odors B, C, and D, which were normally separated by several seconds in real-time, were sequentially reactivated within individual odor presentations. On odor B trials (Fig. 4a–c), we found that the representation of B was significantly predicted before the odor was present ($-200$ ms to $50$ ms window) and maintained (significantly above that of other odors) until it peaked in the 200–450 ms period. Of particular interest is that, while still presented with odor B, the representation of upcoming items in the sequence (C and D) were then activated in the subsequent time windows. In fact, the probability of B significantly declined as the probability of C and D increased (positive slope for "$P_C$ – $P_B$" difference across windows: $t_{(6)} = 4.914$, $P = 0.00796$; positive slope for "$P_D$ – $P_B$" difference: $t_{(6)} = 5.069$, $p = 0.00357$). Importantly, a similar pattern was observed on odor C trials (Fig. 4d–f). In fact, not only was peak decoding of odor C occurring in the 200–450 ms window and followed by the upcoming item in the sequence (odor D; positive slope for $P_D$ – $P_C$; $t_{(6)} = 4.086$, $p = 0.0064$), the representation of odor B was also activated before the presentation of odor C (positive slope for $P_C$ – $P_B$; $t_{(6)} = 9.078$, $p = 0.0008$).

Similar sequential decoding peaks of B, C, and D were also observed on odor D trials (Supplementary Fig. 4), further supporting the consistency of the pattern. However, it is important to note that the representation of odor A was not strongly decoded throughout presentations of odors B, C, or D. This is consistent with the fact that the first item of the sequence was always odor A (whereas any odor could be presented in subsequent sequence positions) and thus the discriminative part of the sequence started at the second sequence position. Importantly, the pre-trial predictive coding above (decoding of B before odor B or C trials) was significantly stronger before correct than incorrect responses (Kolmogorov–Smirnov $D = 0.3683$, $p = 0.0175$; $D = 0.3790$, $p = 0.028$ when downsampled), suggesting it is critically linked to order memory judgments. As with the Bayesian model, this effect was tested on OutSeq trials for adequate sampling of correct and incorrect responses, using the 250 ms window before port entry to avoid the potential influence of the OutSeq stimulus (see "Methods" section). Importantly, we also ruled out the possibility that these findings were influenced by replay events associated with sharp-wave ripples (SWRs), which were exceedingly rare in the windows of interest (see "Control analyses" in the "Methods" section). Collectively, these findings provide compelling evidence of a theta-associated (i.e., not SWR-associated) form of sequence reactivation capturing the sequential relationships among discontiguous nonspatial events, which may reflect a schematic representation of the sequence that was retrieved and played forward within each event presentations to guide behavioral decisions.

**Nonspatial event representations are sequentially organized within theta cycles and precess across successive cycles.** We then examined whether these sequential representations of nonspatial items could also be compressed within individual theta cycles, a phenomenon originally reported in place cell studies (theta sequences)[19–23]. During spatial navigation, theta sequences are thought to represent a segment of trajectory in space: within a single theta cycle, the animal's preceding location tends to be represented in the descending phase, its current location at the trough, and its subsequent location in the ascending phase. Here, we tested the hypothesis that a similar organization extends to sequences of nonspatial stimuli that are conceptually linked but normally separated by seconds (Fig. 5a, top). Since decoding individual theta cycles is beyond the temporal resolution of the CNN model from the previous section, we used a simpler multinomial logistic regression model to quantify odor decoding at faster timescales (see "Theta sequence decoding analyses" in "Methods" section). Note that this model is more appropriate for decoding categorical variables, in this case, the different odors presented, than the Bayesian model from the first section.

Our first analysis focused on the first theta cycle beginning 100 ms after port entry (cycle 1), to capture the period when hippocampal processing should be reflecting the currently presented odor. The model was trained using the spiking activity of the ensemble during the trough of that theta cycle (120–240 degrees; CA1 pyramidal layer theta; InSeq correct trials only) and was then used to decode odor information during the entire cycle (ascending phase, trough and descending phase). Consistent with evidence of theta sequences during spatial navigation, we found that information about past, present, and future stimuli were differentially represented across the descending phase, trough, and ascending phase of theta (Fig. 5a). More specifically, the present (currently presented) stimulus was most strongly represented at the trough ($t_{\text{DescendingVsTrough}\,(294)} = 10.2727$, $p < 0.0001$; $t_{\text{TroughVsAscending}\,(294)} = 9.4514$, $p < 0.0001$), the descending phase showed significantly higher decoding of the past stimulus than of the future stimulus ($t_{\text{PastVsFuture}\,(294)} = 2.2098$, $p = 0.0189$), but the opposite pattern was observed in the ascending phase (future > past; $t_{\text{PastVsFuture}\,(294)} = 2.8302$, $p = 0.0025$; Ascending/Descending × Past/Future ANOVA $F_{(1,294)} = 12.16$, $p = 0.0006$). To help visualize this pattern, we also ran the model on each neuron independently to capture how the coding of individual neurons varied across theta phases. Importantly, since the decoding of the present odor was particularly strong across phases (consistent with the ensemble results in Fig. 5a), we z-normalized past, present, and future decodings separately to highlight their respective dynamics across phases (Fig. 5b). This approach revealed that the sequential organization within the cycle could also be observed at the single neuron level: decoding of past, present, and future stimuli were stronger in the descending, trough, and ascending phases, respectively (Fig. 5b, bottom; significant diagonal relative to shuffled data; $p < 0.0001$; see "Methods" section). These findings indicate that the representation of past, present, and future nonspatial events, which are separated by several seconds in real-time, can be compressed within a theta cycle.

We then examined whether this sequential information contained within theta cycles is associated with decision accuracy and whether it flexibly incorporates trial-specific contingencies. To do so, we used the same model training as above (ensemble activity in trough of cycle 1; correct InSeq trials) but applied model testing to other trial types and cycles. As in previous sections, we examined the effect of accuracy by comparing decodings between correct and incorrect OutSeq trials in the time window preceding port entry (aggregating data over the two pre-stimulus cycles corresponding to the $-250$ ms to 0 ms window; Fig. 5c). Consistent with the prediction that the ascending phase is linked with the processing of upcoming information, we found that decoding of the expected (InSeq) stimulus in that phase was significantly higher on correct than incorrect trials ($F_{\text{CorrectVsIncorrect} \times \text{Phases}\,(2,88)} = 3.717$, $p = 0.028$; $t_{\text{Ascending}\,(44)} = 2.683$, Bonferonni-corrected $p = 0.0027$; downsampled comparison: $t_{\text{Ascending}\,(30)} = 2.1435$, $p = 0.020$). We assessed the flexibility of this form of theta sequences by determining whether

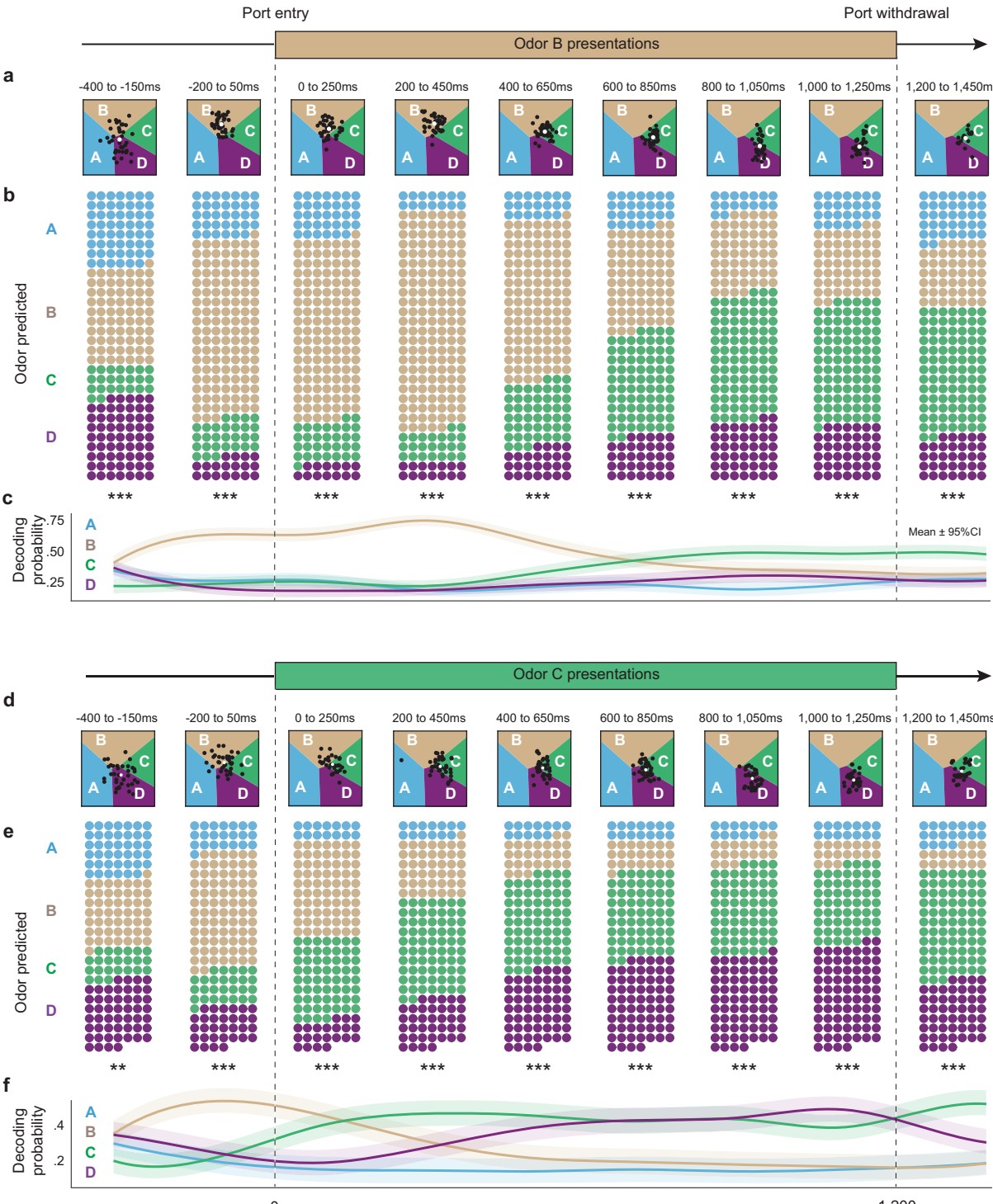

**Fig. 4 The neural representation of sequences of discontiguous events is reactivated within individual events.** A deep learning neural decoding approach was used to quantify the decoding probability of each odor representation in 250 ms windows (correct InSeq trials only). **a** Decoding dynamics during odor B presentations in example subject. Regions of latent space are color-coded according to the odor of highest probability (odor A = sky blue, B = brown, C = green, D = purple), with boundaries indicating equal probability between two odors. Each black dot indicates the latent representation of a trial in that time window, with its position representing the decoded probabilities for each odor (cluster centroids shown as white circles). **b**, **c** Group decoding dynamics during odor B presentations. Since latent space coordinates were specific to each rat's neuronal ensemble, trial data were pooled across subjects according to each trial's highest probability value. Specifically, for each time window, trials (represented by circles; $n = 210$) were categorized and color-coded by their decoded odor of highest probability, with the number of circles in each window reflecting the decoded probability for each odor (**b**). The same data are shown in a summary line plot to highlight the decoding probability dynamics within trials (**c**). **d** Decoding dynamics during odor C presentations in the same example subject. **e**, **f** Group decoding dynamics during odor C presentations. The same approach as above was used to aggregate trial data across subjects ($n = 165$ trials). Data in **b**, **e** show significant chi-square tests in all time windows (df = 3; **$p < 0.005$; ***$p < 0.0001$). Summary line plots in **c**, **f** depict decoding probability for each odor (mean ± 95% multinomial CI; smoothed using cubic splines).

they differed between InSeq and OutSeq trials. More specifically, we compared their decoding of past, present, and future items across the three phases of the first cycle (cycle 1; Fig. 5d). If the same pattern were observed between InSeq and OutSeq trials, it would indicate that these theta sequences rigidly reflect the most common sequence of items (InSeq) even if a different (OutSeq) stimulus is actually presented (or that they simply reflect information about sequence positions). Instead, we observed that the decoding was significantly different, but only in the trough ($F_{\text{InSeqVsOutSeq} \times \text{Phases (2,650)}} = 6.779$, $p = 0.001$; $t_{\text{Trough (325)}} = 3.819$, Bonferonni-corrected $p < 0.0001$). The significantly lower decoding in the trough indicates the activity on OutSeq trials did not simply represent the same information as on InSeq trials (i.e., it reflected the OutSeq nature of the trial), whereas the lack of a significant difference on the descending and ascending phases suggests the same past and future InSeq stimuli were coded on InSeq and OutSeq trials which is consistent with task contingencies (i.e., items preceding and following OutSeq items are always InSeq). Taken together, these results suggest that this form of theta sequence information is important to perform accurate order judgments and flexibly captures trial-specific features associated with task demands.

Finally, we investigated how the information represented varied across a series of equidistant theta cycles within individual odor presentations, ranging from a pre-stimulus cycle to a cycle near the end of the odor presentation (cycle 5). First, we examined how the ensemble decoding of past, present, and future stimuli varied across cycles. To do so, we used the same model training period as above but now tested the model across the four cycles. Notably, we $z$-normalized the decoding values of past, present, and future stimuli separately to highlight their respective magnitude and dynamics across phases and cycles (Fig. 6a), though the same pattern of statistical significance was observed without normalization. When focusing on the overall pattern on each cycle (collapsing across phase; Fig. 6b), we observed a structure consistent with the sequential reactivation reported in the previous section. More specifically, we found that decoding of past, present and future stimuli exhibited significantly different distributions across cycles ($F_{\text{Stimulus} \times \text{Cycles (6,10656)}} = 64.288$, $p < 0.0001$): past stimulus decoding increased toward the pre-stimulus cycle ($F_{\text{LinearTrend (1,888)}} = 89.57$, $p < 0.0001$), present stimulus decoding peaked in cycle 1 ($F_{\text{QuadraticTrend (2,888)}} = 48.34$, $p < 0.0001$), and future stimulus decoding increased toward the last cycle ($F_{\text{LinearTrend (1,888)}} = 103.5$, $p < 0.0001$). Second, we examined whether the information coded by individual neurons precessed across cycles. To do so, we adapted the single-cell visualization method used in Fig. 5b, in which the model was run on each neuron independently. Consistent with the canonical view of theta phase precession from the place cell literature[20,21] (Fig. 6c), we found that the peak decoding of the present stimulus shifts from a late to an earlier phase within trials (Fig. 6d; $r = -0.377$, $p = 0.007$; $p = 0.002$ when examined using a permutation approach to maintain the overall strength of decodings within trials but disturb their phase). We also used standard analyses to determine whether the spiking activity of individual neurons precessed within trials. Although the complexity of our design is not well-suited to quantify this (our use of many odors makes odor-specific coding more graded and limits the number of trials on each), we did observe a proportion of neurons with significant phase precession during the trial period (26.1% of neurons when collapsing data across BCD trials; 17.7%, 14.1%, and 16.8% when examining B, C, and D trials separately; see examples in Supplementary Fig. 5). These results suggest that the ensemble representations of past, present, and future stimuli are sequentially organized within individual stimulus presentations, a pattern that parallels the sequence reactivation observed with the CNN model in the previous section, and that the information represented in individual neurons precesses across theta cycles.

## Discussion

In this study, we leveraged complex behavioral and statistical machine learning approaches to discover forms of sequential organization in hippocampal ensemble activity supporting the memory for sequences of events, a capacity known to critically depend on the hippocampus[31–33]. We report that neurons with time-locked firing fields (time cells) were observed across a series of discrete nonspatial events distributed over several seconds, and that this temporal signal was linked with correct order memory judgments. We also found that CA1 ensembles simultaneously and sequentially differentiated distinct types of trial-specific information within stimulus presentations, including the stimulus presented, its temporal order, and whether the animal correctly identified the trial type. In addition, despite the stimuli being separated by several seconds in real-time, we discovered that hippocampal ensembles reactivated the corresponding sequential relationships among past, present, and future stimuli within individual stimulus presentations and that these sequential representations can be compressed within a single theta cycle. We also found that the decoded information precessed in theta phase across successive cycles within individual trials. Collectively, these results strongly suggest that encoding, preserving, and predicting event sequences is fundamental to hippocampal function.

Sequential firing fields (time cells) have previously been reported during individual stimulus presentations, inter-stimulus intervals, and across contiguous stimuli and responses[11–15]. Considerable evidence indicates this form of coding can provide a strong temporal signal within such task events[14,15]. Here we offer direct evidence that hippocampal sequential firing fields provide significant temporal information across a sequence of dis-contiguous events unfolding over several seconds, and that this form of coding is important to correctly remember the order of events. These findings lend support to theoretical models proposing that this form of coding supports our ability to organize our experiences in time[4,27,28,34], a characteristic feature of epi-sodic memory[1–4]. In light of recent evidence that the lateral entorhinal cortex, a region with a strong anatomical and functional relationship with the hippocampus, provides a robust temporal signal across a longer timescale (several minutes)[35,36], our results suggest that different aspects of the temporal context of episodic memories may be represented across medial temporal lobe structures.

Our unique behavioral and analytical approaches allowed us to extend previous demonstrations of conjunctive or stimulus-specific odor coding in hippocampal neurons, including indivi-dual neurons responding to a specific odor in a specific sequence position[26] or to a specific moment in time during a specific stimulus[14]. Here we extend these previous findings by discovering that the representations of individual stimuli also contain infor-mation about their sequential relationships. In fact, we demon-strate a temporal lag effect in which the temporal distance between stimuli (in terms of sequence position) was reflected by the degree of dissimilarity of their neural activity, an effect that was observed using different metrics (temporal coding accuracy in Fig. 2c, distance between clusters in the low-dimensional representation of the neural activity in Fig. 3b). This effect pro-vides a potential neural mechanism for our ability to judge the relative temporal distance and order of discrete experiences that is consistent with computational models of the representations of the temporal context of events[27,28]. We also provide a significant extension to our previous demonstration of individual neurons

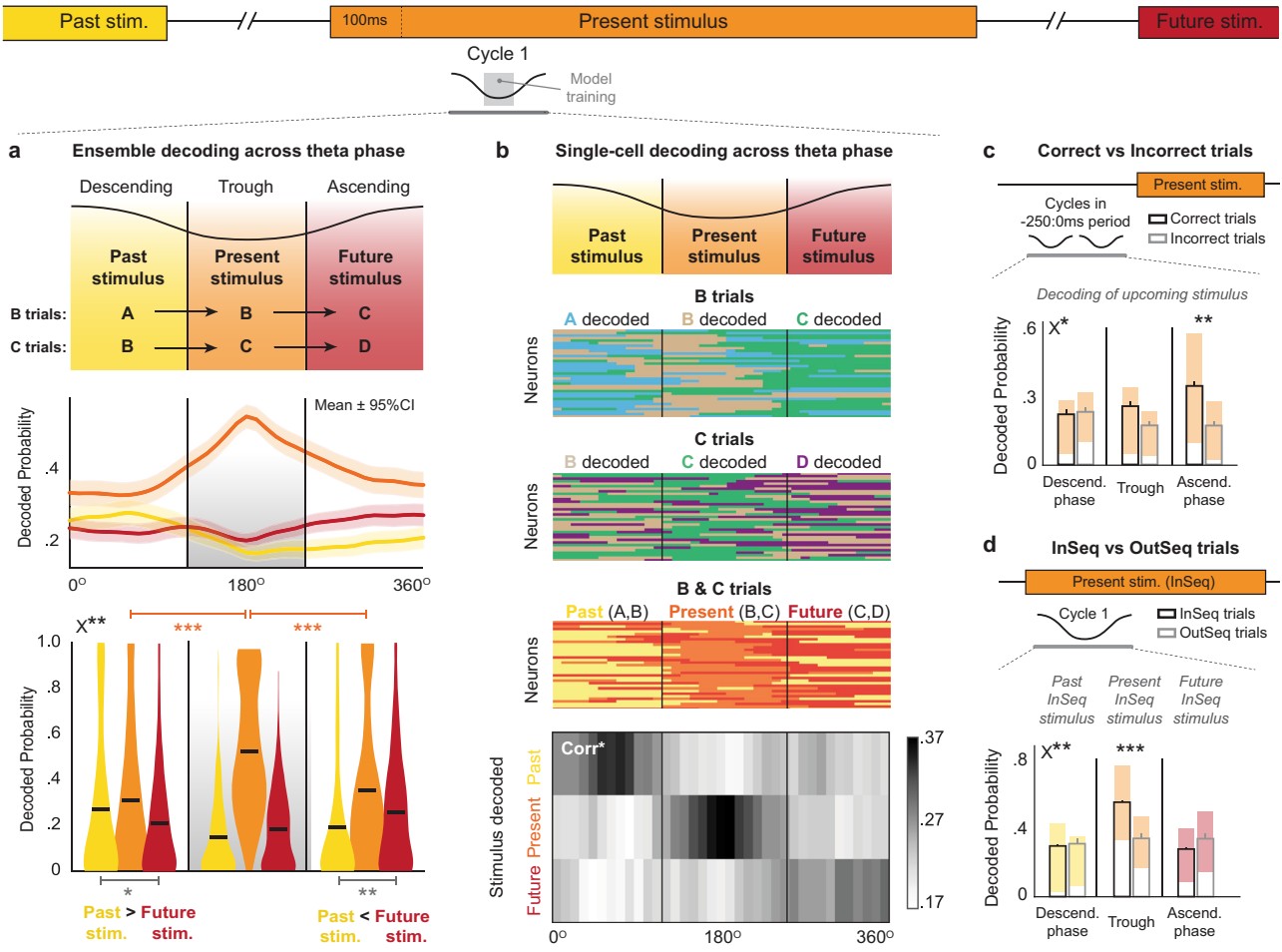

**Fig. 5 The representations of past, present and future nonspatial events are sequentially organized within theta cycles and associated with decision accuracy and trial-specific contingencies.** A multinomial logistic regression model was used to quantify odor decoding within theta cycles. **a** Ensemble decoding during cycle 1 (first cycle beginning 100 ms after port entry). Top, Hypothesized pattern of theta sequence organization. Middle, decoded probability dynamics for past (yellow), present (orange), and future (red) stimuli ($n = 295$ trials, pooled across odor B & C trials and subjects; mean ± 95% CI). Bottom, The descending phase showed significantly higher decoding of past than of future stimuli, whereas the opposite pattern was found in the ascending phase (phase × stimulus $F_{(1,294)} = 12.16$, $p = 0.0006$; $t_{Descending(294)} = 2.2098$, $p = 0.0189$; $t_{Ascending(294)} = 2.8302$, $p = 0.0025$). Decoding of present stimuli was stronger during the trough than either the descending ($t_{(294)} = 10.2727$; $p < 0.0001$) or ascending ($t_{(294)} = 9.4514$; $p < 0.0001$) phases. Horizontal black lines depict means; gray shading indicates model training period. **b** Single-cell decoding during cycle 1. Top, hypothesized pattern. Middle three panels, decoding of past, present, and future stimuli in same example neurons across trials of odor B, C, and B & C combined. Color indicates odor with highest z-scored probability per 10° bin (Odor A = sky blue, B = brown, C = green, D = purple). Bottom, Proportion of phase-modulated neurons coding for past, present, and future stimuli in each 10° bin (B & C trials; Corr*, significant stimulus-phase correlation determined by permutations). **c** Theta sequence information was associated with decision accuracy. Ensemble decoding accuracy of upcoming stimulus was higher on correct ($n = 60$) than incorrect ($n = 30$) trials during ascending phase (averaged over two pre-trial cycles; performance × phase $F_{(2,88)} = 3.717$, $p = 0.0282$; $t_{Ascending(44)} = 2.683$, Bonferonni-corrected $p = 0.0027$). **d** Theta sequence information flexibly reflected trial-specific contingencies. Using decoding of InSeq stimuli as reference, InSeq ($n = 297$) and OutSeq ($n = 30$) trials were shown to differ in content during the trough of cycle 1 (capturing the OutSeq nature of trials) but not in the descending and ascending phases (InSeq/OutSeq × phase; $F_{(2,650)} = 6.779$, $p = 0.001$; $t_{Trough(325)} = 3.819$, Bonferroni-corrected $p < 0.0001$). Bars graphs in **c**, **d** depict mean ± SEM (B & C trials, pooled across subjects), with shaded regions reflecting Q1–Q3 range (median as white line). All $t$-tests two-tailed (*$p < 0.05$; **$p < 0.005$; ***$p < 0.0001$; uncorrected in panel **a** because defined a priori). X* and X** denote significant ANOVA interactions ($p < 0.05$ and $p < 0.005$, respectively).

firing differentially to InSeq and OutSeq items[26], by revealing that this differentiation reflects more than just a match/mismatch signal. In fact, since the lag effect above was observed for both InSeq and OutSeq items, it indicates that the differential activity observed between InSeq and OutSeq items also captures "how far off" an OutSeq item is from its expected sequence position. This representation of the temporal lag among events, and the simultaneously observed sequential coding properties above, suggest a strong "multiplexing" of different forms of temporal information in hippocampal ensemble activity, akin to what has been reported in the spatial domain[37].

Our findings support and extend recent evidence of theta-associated predictive and temporal coding properties in hippocampal ensembles. First, a previous study has reported theta-associated sequential representations that reflected the length of upcoming spatial trajectories[22]. These representations are thought to play a key role in planning and decision-making by maintaining an online representation of the planned trajectory toward the current goal. Importantly, the present study provides evidence that such predictive coding extends to nonspatial information that is not directly perceptible and is directly linked with correct sequence memory judgments. Notably, these findings are also

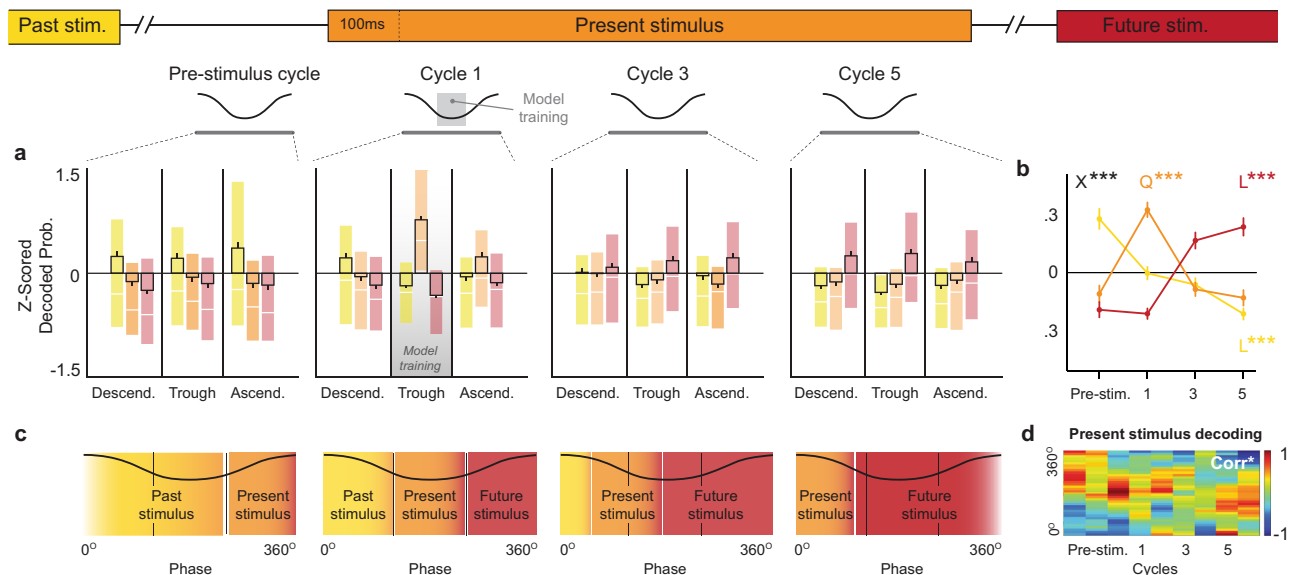

**Fig. 6 The representations of nonspatial events display sequential decoding peaks and phase precession across theta cycles. a, b** Ensemble decoding showing sequential peaks of past, present, and future stimulus coding across cycles consistent with sequence reactivation in Fig. 4 ($n = 297$; B & C trials combined and pooled across rats). **a** Decoding probabilities across phases and cycles. Past (yellow), present (orange), and future (red) stimulus probabilities were separately z-normalized using their data from all plots to highlight their respective dynamics across phases and cycles. **b** Summary and quantification of dynamics across cycles shown in **a**, collapsed across phase. The decoding of past, present, and future stimuli exhibited significantly different distributions across cycles (stimulus × phase $F_{(6,10656)} = 64.288$, $p < 0.0001$). Past stimulus decoding decreased across cycles (linear trend $F_{(1,888)} = 89.57$, $p < 0.0001$), whereas future stimulus decoding increased (linear trend $F_{(1,888)} = 103.5$, $p < 0.0001$), and present stimulus decoding showed a curvilinear relationship peaking in cycle 1 (quadratic trend $F_{(2,888)} = 48.34$, $p < 0.0001$). **c** Hypothesized decoding pattern based on demonstrations of theta phase precession from the place cell literature. **d** Theta phase precession of present stimulus decoding across cycles. Data shown represents the sum of z-scored decodings of all neurons ($n = 251$; pooled across animals, normalized to cycle maximum; see "Methods" section). Peak decoding of present stimulus significantly varied in phase across cycles (Corr*; $r = -0.377$, $p = 0.007$), from a late (ascending) phase to an early (descending) phase. Data in **a**, **b** depict mean ± SEM (B & C trials, pooled across subjects). Shaded regions in **a** depict Q1–Q3 range (median denoted by white line). L*** and Q*** indicate significant linear and quadratic trends ($p < 0.0001$). X*** denotes significant interaction by two-way ANOVA ($p < 0.0001$).

compatible with the view that the hippocampus plays a key role in representing task-related schemas[38], in this case, the sequential relationships among those events, to guide behavioral decisions. Second, our findings are consistent with a recent study that used a conditional discrimination task with overlapping pairs of stimuli (a tone and an odor) to demonstrate theta phase precession for nonspatial information in hippocampal neurons[14]. In that study, the authors also used reconstructed time estimates to show that immediately preceding and upcoming time points (on the order of a few hundred ms) could be represented within a theta cycle. Our study extends these findings by suggesting a higher level of abstraction for nonspatial theta sequences—that the sequential relationships among past, present, and future events (separated by seconds in real-time) can also be compressed within a theta cycle.

Our findings also raise a number of important questions to address in future studies. As we demonstrated, the sequence coding properties reported here were associated with correct order memory judgments. How can such representations in the hippocampus directly influence decisions and responses? It has been proposed that theta oscillations may play an essential role in allowing prospective hippocampal representations to facilitate decision-related processing in downstream structures[9,22,37]. For instance, it is well established that neural activity in regions associated with planning behavior, including the prefrontal cortex and striatum, is significantly entrained to the hippocampal theta rhythm[39–41]. This proposal is consistent with work using other temporal memory tasks showing that, in addition to the hippocampus, the prefrontal cortex is also strongly engaged during task performance in both rodents and humans[32,42–44]. However, the specific prefrontal mechanisms remain to be demonstrated. Finally, it remains to be determined

whether the forms of sequence coding reported here are present in other medial temporal lobe structures, and how these "online" theta-associated representations interact with "offline" SWR-associated representations related to upcoming behavior or goals observed during reward consumption, quiescence, or sleep[16,17].

## Methods

**Animals**. Subjects were five male Long–Evans rats, weighing ~350 g at the beginning of the experiment. They were individually housed and maintained on a 12 h light/dark cycle. Rats had ad libitum access to food, but access to water was limited to 2–10 min each day, depending on how much water they received as a reward during behavioral training (3–6 ml). On weekends, rats received full access to water for ≥12 h to ensure adequate overall hydration. Hydration levels were monitored daily. All procedures were conducted in accordance with the Institutional Animal Care and Use Committee (Boston University and the University of California, Irvine).

**Equipment and Stimuli**. The apparatus consisted of a linear track (length, 150 cm; width, 9 cm), with walls angled outward (30 degrees from vertical; height, 40 cm). An odor port, located on one end of the track, was equipped with photobeam sensors to precisely detect nose entries and was connected to an automated odor delivery system capable of repeated deliveries of multiple distinct odors. Two water ports were used for reward delivery: one located under the odor port (used to reward correct order judgments), the other at the opposite end of the track (used to encourage rats to leave the odor port area before the presentation of the next sequence). Timing boards (Plexon) and digital input/output boards (National Instruments) were used to measure response times and control the hardware. All aspects of the task were automated using custom Matlab scripts (MathWorks). A 96-channel Multichannel Acquisition Processor (MAP; Plexon) was used to interface with the hardware in real-time and record the behavioral and electrophysiological data. Odor stimuli consisted of synthetic food extracts contained in glass jars (A, lemon; B, rum; C, anise; D, vanilla; E, banana) that were volatilized with desiccated, charcoal-filtered air (flow rate, 2 L/min). To prevent cross-contamination, separate Teflon tubing lines were used for each odor, which converged

into a single line at the bottom of the odor port. In addition, an air vacuum located at the top of the odor port provided constant negative pressure to quickly evacuate odor traces. Readings from a volatile organic compound detector confirmed that odors were cleared from the port 500–750 ms after odor delivery (inter-odor intervals were limited by software to ≥800 ms).

**Behavior**. Naive rats were initially trained to nosepoke and reliably hold their nose for 1.2 s in the odor port for a water reward. Odor sequences of increasing length were then introduced in successive stages (A, AB, ABC, ABCD, and ABCDE) upon reaching the behavioral criterion of 80% correct over three sessions per training stage. In each stage, rats were trained to correctly identify each presented item as either InSeq (by holding their nosepoke response for at least 1.2 s to receive a water reward) or OutSeq (by withdrawing their nose before 1.2 s to receive a reward). Note that OutSeq items could be presented in any sequence position except the first (i.e., sequences always began with odor A, though odor A could also be presented later in the sequence as an OutSeq item). After reaching criterion performance on the five-item sequence (>80% correct on both InSeq and OutSeq items), rats underwent surgery for microdrive implantation.

Only one sequence was used during training (ABCDE; same odors and order across rats) and the same sequence was used in the recorded session. Although we have counterbalanced sequences in behavioral studies using the same task[25,45], we chose to match the sequences across animals in this recording study to keep the neural coding as consistent as possible across rats. Note that, given the stepwise training, animals had more exposure to the earlier odors in the sequence in their lifetime, though all odors were highly familiar to them (they received hundreds of presentations of each during training). Therefore, differences in decoding levels are viewed as primarily related to sampling idiosyncrasies of the recorded session (the number of trials for each odor and the coding properties of each ensemble).

**Surgery**. Rats received a preoperative injection of the analgesic buprenorphine (0.02 mg/kg, s.c.) ~10 min before induction of anesthesia. General anesthesia was induced using isoflurane (induction: 4%; maintenance: 1–2%) mixed with oxygen (800 mL/min). After being placed in the stereotaxic apparatus, rats were administered glycopyrrulate (0.5 mg/kg, s.c.) to help prevent respiratory difficulties. A protective ophthalmic ointment was then applied to their eyes and their scalp was locally anesthetized with marcaine (7.5 mg/mL, 0.5 ml, s.c.). Body temperature was monitored and maintained throughout surgery and a Ringer's solution with 5% dextrose was periodically administered to maintain hydration (total volume of 5 mL, s.c.). The skull was exposed following a midline incision and adjustments were made to ensure the skull was level. Six support screws (four titanium, two stainless steel) and a ground screw (stainless steel; positioned over the cerebellum) were anchored to the skull. A piece of skull ~3 mm in diameter (centered on coordinates: −4.0 mm AP, 3.5 mm ML) was removed over the left hippocampus. Quickly after the dura was carefully removed, the base of the microdrive was lowered onto the exposed cortex, the cavity was filled with Kwik-Sil (World Precision Instruments), the ground wire was connected, and the microdrive was secured to the support skull screws with dental cement. Each tetrode was then advanced ~900 μm into the brain. Finally, the incision is sutured and dressed with Neosporin, and rats were returned to a clean cage, where they were monitored until they awoke from anesthesia. One day following surgery, rats were given an analgesic (flunixin, 2.5 mg/kg, s.c.) and a topical antibiotic (Neosporin) was reapplied to the incision site.

**Electrophysiological recordings**. Each chronically implanted custom microdrive contained 21 independently drivable tetrodes. Each tetrode consisted of four twisted nichrome wire (13 μm in diameter; California Fine Wire) and gold-plated to achieve final tip impedance of ~250 kΩ (measured at 1 kHz). Following a surgical recovery period of five days, tetrodes were slowly advanced over a period of ~3 weeks while monitoring established electrophysiological signatures of the CA1 pyramidal cell layer (e.g., sharp waves, ripples, and theta amplitude). Voltage signals recorded from the tetrode tips were referenced to a ground screw positioned over the cerebellum, and differentially filtered for single-unit activity (154 Hz to 8.8 kHz) and LFP (1.5–400 Hz). The neural signals were then amplified (10,000–32,000× for units, 1000× for LFP), digitized (40 kHz for units, 1 kHz for LFP), and recorded to disk with the data acquisition system (MAP, Plexon). Action potentials from individual neurons were manually isolated offline using a combination of standard waveform features across the four channels of each tetrode (Offline Sorter, Plexon). Proper isolation was verified using interspike interval distributions for each isolated unit (assuming a minimum refractory period of 1 ms) and cross-correlograms for each pair of simultaneously recorded units on the same tetrode. To confirm recording sites, current was passed through the electrodes before perfusion (0.9% PBS followed by 4% paraformaldehyde) to produce small marking lesions, which were subsequently localized on Nissl-stained tissue slices.

**Data inclusion, sampling, and pooling**. We took a rigorous and conservative approach to data inclusion, sampling, and pooling to maximize the reproducibility of the findings. First, we only used data from one session per animal to avoid oversampling neurons and trials (only one session was recorded per day). In other words, we did not combine an animal's neurons or trials across daily sessions to

artificially increase statistical power. Second, to balance the flexibility, power, and rigor of the analyses, our primary statistical approach was to treat decoded probabilities from each trial as individual samples, and the trial data were then pooled across subjects to perform statistical comparisons at the group level. Whenever possible, we performed key comparisons using a "within-ensemble" design (e.g., how the pattern of decoding probabilities from each subject's ensemble varied across time periods or trial types) using omnibus tests (e.g., repeated-measures ANOVAs) followed by posthoc tests controlling for the number of comparisons performed. Third, we took a conservative approach to cell inclusion criteria to avoid unintentionally biasing the analyses. Briefly, we only excluded neurons if their firing rate values would create problems with an analysis (e.g., a firing rate of zero across all trials of a specific type would leave some calculations undefined). Thus, instead of filtering neurons based on their degree of informativeness, we took a more systematic approach and let the models determine which neurons were informative or uninformative. Note that the strength of our findings either remained unchanged or significantly improved when excluding uninformative neurons (see Supplementary Fig. 6 and "Control analyses" section below). Fourth, upon review of the behavioral data during pre-processing, we identified rare instances where the rat's behavior was scored incorrectly due to extremely brief entries into the port where the system only registered a rat's port entry and not its withdrawal. These trials were removed and not considered as part of the analyses reported here. Post-processing trial and neuron counts are provided in Supplementary Tables 1 and 2. Finally, note that a small fraction of tetrodes (<10%) may have been located within the borders of neighboring subregion CA2. Since that number of neurons is too small to properly quantify ensemble dynamics and that there was no clear evidence their single-cell coding properties differed from the rest, these cells were included in the ensemble analyses.

**Peri-stimulus time histograms and correlation analyses**. The activity of individual neurons during odor presentations was visualized using peri-stimulus time histograms (PSTHs) averaging data from all correct InSeq trials from the session. Each row represents the mean normalized firing rate from a single neuron across trials of a specific type. More precisely, for each neuron, the activity was binned into 1 ms bins, smoothed across bins using a gaussian filter (150 ms for individual odor PSTHs in Fig. 2a, 250 ms for full sequence PSTHs in Fig. 2d), collapsed across trials to obtain the mean firing rate per bin, and finally normalized to its peak rate within the row. PSTHs for individual odor presentations (e.g., Fig. 2a) aggregated neurons across all subjects to easily visualize the activity from all active neurons. However, to be conservative, PSTHs for full sequences (e.g., Fig. 2d) only displayed neurons from a single subject to acknowledge potential differences in the pacing of trials across subjects. For each PSTH, neurons that did not fire in any of the corresponding trials were excluded (though such neurons may be included in another PSTH).

The similarity among odor PSTHs was quantified using a simple correlational approach (Supplementary Fig. 2a). First, to balance the sampling, we used the same number of trials for each odor type (first 14 correct InSeq trials for odors A, B, C, and D; 56 trials in total). Then, we essentially produced a PSTH for each trial (neurons aggregated across subjects; firing rate in 1 ms bins, gaussian smoothed and normalized) and correlated all pairs of trial-specific PSTHs (within and across odors). Correlation coefficients ($r$) for the different combinations of trial types (AvsA, AvsB, AvsC, etc) were statistically compared using one-way ANOVAs. Dunnett's posthoc tests were then used to directly compare "same odor" vs "different odor" pairs (e.g., comparing AvsA with AvsB, AvsC, and AvsD) while controlling for the number of comparisons performed. Statistical significance was determined using $p < 0.05$.

**Temporal coding analyses**. The objective of this approach was to quantify the degree of temporal information contained in sequential firing fields from each animal's neuronal ensemble, by assessing the degree to which time can be accurately decoded from the ensemble activity. To do so, we adapted a memoryless Bayesian decoding model from a recent study examining temporal coding during sampling of olfactory and auditory stimuli[14]. To match the unbiased approach we used with the other models, temporal coding analyses included all active neurons (i.e., neurons that fired at least one spike during any of the odor presentations). In other words, neurons were not filtered by their coding properties to avoid potentially biasing the analyses. Notably, the same pattern of results was observed when non-informative neurons were excluded (see "Control analyses" below; compare Fig. 2 with Supplementary Fig. 6). In addition, significant reconstructed time accuracy was also observed when using a more conservative training/test validation approach (i.e., 50:50 splits instead of leave-one-out cross-validation; also in "Control analyses" below; Supplementary Fig. 7). Unless specified otherwise, an animal's ensemble decoding accuracy values for each trial (i.e., each odor presentation; see "accuracy of reconstructed time" below) were treated as individual samples, and the trial data were then pooled across subjects to be analyzed at the group level. Statistical significance was determined using $p < 0.05$.

*Bayesian model for time reconstruction*. By assuming the ensemble spiking activity has a Poisson distribution and the spiking activity of each neuron is independent, we can use the spiking activity in a specific time bin (actual time) to calculate the

probability of time (reconstructed time) in Fig. 2b as follows:

$$P(time|spikes, odor) = \frac{P(spikes|time, odor).P(time|odor)}{P(spikes|odor)} \quad (1)$$

$$P(time|spikes, odor) = C.P(time|odor) \prod_{i=1}^{N} \frac{[\tau f_i(time, odor)]^{n_i}}{n_i!} \exp\left[-\tau \sum_{i=1}^{N} f_i(time, odor)\right] \quad (2)$$

In Fig. 2e, we use $P(time, odor|spikes)$ instead. Here, $\tau$ is the length of the time bin (50 ms with step size of 5 ms in Fig. 2b; 1500 ms with step size of 150 ms in Fig. 2e), $f_i(time, odor)$ is the mean firing rate of the $i$-th unit in the time bin, $n_i$ is the number of spikes occurring in the time bin, odor refers to the treatment of individual odors (odors ABCD in Fig. 2b; odors ABCDE in Fig. 2e), $C$ is a normalization factor to make the probability distribution for each time bin (actual time) sum up to 1, and $P(time, odor)$ is a constant since all InSeq trials have the same duration.

Unless otherwise noted, the model was trained using a balanced subset of sequence presentations in which all odors were InSeq, consecutively presented, and correctly identified (using a leave-one-out cross-validation approach). Temporal coding plots (e.g., Fig. 2b) show the posterior probability distribution of reconstructed time for each bin of actual time (calculated separately for each subject, then averaged across subjects).

*Accuracy of reconstructed time.* Decoding accuracy was determined by quantifying the degree of relationship between actual time and reconstructed time estimates for each animal's ensemble separately. To do so, for each trial's reconstructed time matrix, we calculated the correlation across rows and columns of the same index (one correlation value per row-column pair) and the mean of these correlation values represented the accuracy of reconstructed time on that specific trial. Chance levels were determined by calculating the mean reconstructed time accuracy across 1000 random permutations of the time factor in the $f_i(time, odor)$ matrix. For accuracy comparisons across odors (Supplementary Fig. 2b), pooled trial data were analyzed using one-way ANOVAs followed by Dunnett's posthoc tests (comparing "same" vs "different" odors while controlling for the number of comparisons). Comparisons with chance levels were performed using one-sample $t$-tests (two-tailed) using pooled trial data for individual odors (Fig. 2b; using the same number of trials for odors A–D) but separately for each subject for the full sequence (because of variations in the timing of odor presentations across subjects; Fig. 2e).

*Lag analysis.* For the lag analysis, the model was tested on trial types not included in the training set. For lags of 1–3 (positive or negative) the model was trained using the proper sequence position of a given odor, but tested on the other sequence positions (e.g., training during B in ABCDE, but decoding during C, D, or E). For lags of 0, the model was tested on a subset of non-consecutive InSeq trials of comparable size. Statistical comparisons across lags were performed using a one-way ANOVAs (using pooled trial data as samples), followed up by linear trend analyses and pairwise comparisons using Dunnett's posthoc tests (comparing each lag with a lag of 0 while controlling for the number of comparisons).

*Correct vs incorrect trials.* InSeq mistakes are relatively rare in this paradigm when animals are well trained and tend to occur a few ms before the signal, thus most likely reflect errors of anticipation instead of incorrect decisions[25,45]. Therefore, comparisons between correct and incorrect trials focused on OutSeq trials. More specifically, the model was trained on the full sequence of InSeq odors (as in Fig. 2e) but tested on the 250-ms window preceding port entry on OutSeq trials (to avoid the confounding influence of the OutSeq odor presentation). Only OutSeq trials in sequence positions 2–4 were included to avoid edge effects and maximize alignment across trial types (81 correct vs 33 incorrect trials). OutSeq reconstructed time accuracy was quantified through comparison with the corresponding decoding on InSeq trials (which served as expected values). More specifically, we used a Kullback–Leibler (KL) divergence analysis to compare the shape of the probability distribution from each OutSeq trial with that of the mean probability distribution from InSeq trials. Statistical comparisons between correct and incorrect trials used non-parametric Kolmogorov–Smirnov tests to account for potential non-normality in the distribution of KL divergence values, though the same pattern of results was observed with parametric tests. Similarly, the result was the same when the comparison was downsampled (by repeating the analysis with 1000 permutations using 33 randomly-selected correct trials).

**Latent representation analyses.** The objective of this approach was to visualize and quantify differences in the underlying structure of each animal's ensemble activity at different moments within individual stimulus presentations. To do so, we used an autoencoder, a nonlinear dimensionality reduction method based on neural networks[46,47], to identify a low-dimensional latent representation of the spike activity data in 250 ms time windows. Note that the autoencoder was run on each subject's data separately to obtain their unique latent representations, but classification accuracy was examined by pooling trial data across all subjects. We used an unbiased approach and included the data from all recorded neurons (neurons were not filtered by their coding properties or firing rate thresholds).

*Autoencoder model.* Data from each trial were divided into 250-ms time windows, aligned to either port entry or withdrawal (see below), and the autoencoder was trained on the data using a sliding window approach (100 ms sub-window, 10 ms steps; 16 data points per window on each trial). Briefly, the model consisted of an input layer (the original neural data), an encoder portion (two layers with 500 nodes each), a "bottleneck" layer (one layer with two nodes), a decoder portion (two layers with 500 nodes each), and an output layer (the reconstructed neural data; see model architecture in Supplementary Fig. 8). The 100 ms sub-windows of spike activity data for each neuron constituted the input layer (the size of the input layer corresponded to the animal's number of neurons multiplied by 10, as the activity was further binned into 10-ms increments). The encoder portion of the model then projected the 100 ms sub-windows of spike activity data (the input layer) onto a two-dimensional latent space by passing it through its two layers and the bottleneck layer (with the activation of the two nodes in the bottleneck layer representing the two dimensions). The decoder portion projected the latent space back into the original space by passing the output from the bottleneck layer through its two layers and finally to an output layer whose dimensionality was the same as the input. The activation function at node $i$ of layer $l$ is defined as:

$$h_l(i) = f(\mathbf{w}_{il}^T \mathbf{h}_{l-1}) \quad (3)$$

where $f$ is a pointwise function (see below), $\mathbf{w}_{il}$ is a vector of learnable parameters, $\mathbf{h}_{l-1}$ represents the output of the previous layer, and $T$ the matrix transposition operation. The input layer (observed data) is referred to as $\mathbf{h}_0$. The model was trained to minimize the difference between the original input data and the reconstructed data. To accomplish this, we estimated the parameters $\mathbf{w}_{il}$, for all $i$ and $l$, using stochastic gradient descent[48] with momentum[49] to minimize the mean squared error between the input and output. We used a rectified nonlinear unit for the pointwise function[50,51], $f$, for all layers except the bottleneck layer, where we used a linear function.

*Visualization of latent representations and k-NN classification.* We focused specifically on the four 250 ms windows in which the behavior was matched between InSeq and OutSeq trials (0:250 ms and 250:500 ms relative to port entry, −250:0 ms and 0:250 ms relative to port withdrawal; see Fig. 3). After training the model, we visualized each subject's two-dimensional latent representation for each window and color-coded each data point according to its trial type: the odor presented (stimulus), whether the odor was presented in or out of sequence (temporal order), and whether the rat performed the correct or incorrect response (trial outcome). Then, we used a k-nearest neighbors (k-NN) with $k = 2$ method to determine when the different types of trial-specific information were well separated in the latent space (70% of trials used for model training, 30% used for testing). More specifically, for each trial in the test set, we predicted the stimulus, temporal order, and trial outcome label based on its top two closest neighbors in the training set. Mean classification accuracy was then determined by pooling trial data across subjects. To focus the k-NN classification on the type of information of interest, the k-NN classification of stimulus and temporal order information only included correct trials (to focus on trial-relevant representations) and odors B, C, and D (to focus on odors with comparable discriminability; including odor A, which was strongly differentiated by the model, further enhanced classification accuracy). Similarly, stimulus classification only included InSeq trials. For statistical comparisons, we determined the 95% confidence interval for chance-level classification accuracy for each type of information and time window using 100 permutations. For each permutation, we randomly shuffled the labels corresponding to stimulus, temporal order, and trial outcome and repeated the k-nearest neighbors classification. Accuracy values above the confidence intervals were considered statistically significant. Note that the same comparisons were also statistically significant when using 99% confidence intervals, with the exception of the temporal order effect in the 250–500 ms window.

*Lag analysis.* To determine if temporal order differentiation (i.e., InSeq vs OutSeq) solely reflected a match/mismatch signal or also contained information about their degree of mismatch (e.g., how "far" an OutSeq odor was from its expected sequence position), we quantified the distance among the cluster representations of each trial type (InSeq odors B–E, OutSeq odors B–E in positions 2–5). To do so, we determined the centroid of each trial in the two-dimensional latent variable space (each trial has 16 data points because the 100 ms sub-window is sliding in 10 ms increments), calculated the Mahalanobis distance between each pair of trial-specific centroids, and categorized trial pairs by their lag. For InSeq trials, lags represented the distance between presented odors: Lag 0 (e.g., B vs B), Lag 1 (e.g., B vs C), Lag 2 (e.g., B vs D) and Lag 3 (e.g., B vs E). For OutSeq trials, lags represented the distance (in sequence position) between two OutSeq presentations of a given odor: Lag 0 (e.g., odor B presented in position 3 vs another trial in which B was in position 3), Lag 1 (e.g., odor D presented in position 3 vs in position 2), Lag 2 (e.g., odor E presented in position 2 vs in position 4) and Lag 3 (e.g., odor C presented in position 5 vs in position 2). Notably, identical results were obtained when either the raw data (all 16 data points per trial) were used instead of centroids, or when Euclidean distance was calculated rather than Mahalanobis distance (Mahalanobis distance was used to account for potential correlations between the two latent variables). Statistical comparisons across lags were performed using one-way

ANOVAs, followed up by linear and quadratic trend analyses. Statistical significance was determined using $p < 0.05$.

**Neural decoding analyses**. The objective of this approach was to build a powerful predictive model for decoding the stimulus information represented in the neural activity data using a supervised learning method. More specifically, we used a convolutional neural network, an approach most commonly used for image classification[52] but that can also be used for sequential data[53]. An animal's ensemble decoding probabilities (the probability of A, B, C, and D) per 250 ms window for each trial (i.e., each odor presentation) were treated as individual samples. Statistical analyses were performed on trial data pooled across subjects using a "within-ensemble" approach (e.g., how decoding varied across windows within a specific trial type). Note that we used an unbiased approach and included the data from all recorded neurons (neurons were not filtered by their coding properties or firing rate thresholds). A control analysis was performed to ensure group effects were not simply driven by particularly strong data from any one subject (see "Control analyses" below).

*Convolutional neural network model (CNN)*. For our analysis, the CNN took both spike data and LFP signals from each tetrode as input, and odor labels as its output. The convolution was performed on each tetrode separately (Supplementary Fig. 9a). More specifically, for each tetrode, the continuous LFP signal and spike activity from each unit were combined into one multivariate time series. Filters (small matrices) were then convolved on these time series to produce convolution features. Time averaged features from different tetrodes were concatenated together and fed into subsequent hidden layers with drop-out[54] and finally to the output layer prediction. The model was trained on data from the 150–400 ms time window (relative to port entry) using a variant of stochastic gradient descent[55] following the early stopping rule. As shown in Supplementary Fig. 9b, this model produced higher decoding accuracy compared to the benchmark multinomial logistic regression model (10-fold cross-validation). Moreover, by using the hidden layer that feeds into the output layer as a latent space, our model allows for nonlinear projection of the original data onto a low-dimensional space for visualizing and quantifying the decoding probability of each odor over time. Note that in contrast to the autoencoder model used for the latent representation learning analyses, this projection in the low-dimensional latent space was obtained in a supervised manner by taking the labels (as well as the neural signals) into account. For each trial, the decoding model was then applied to each 250 ms time window (ranging from −400 ms to 1450 ms, relative to port entry) to obtain the corresponding hidden layer vectors.

*Visualization of latent space*. After the latent space was divided into subregions associated with each odor (according to the odor of highest probability), the latent space vectors for each trial could be visualized as a point that moves around in the latent space across different time windows. For easier visualization (Fig. 4a, d), we have further reduced the dimension of the latent space to two using the top two principal components and linearized the boundaries between odors using a simple multinomial logistic regression model. To aggregate decoding results across subjects (Fig. 4b, e), for each time window, we compiled the number of points (trials) in each odor subregion of each subject's latent space. As a result, for each time window, we obtained four clusters of color-coded points representing the aggregate decoding probability of odors A, B, C, and D across all trials (the total number of points remains constant across windows).

*Statistical comparisons*. For statistical comparisons, we determined the 95% confidence intervals of the decoding probabilities in each time window, which follow a multinomial distribution. To be conservative, decoding probabilities with non-overlapping confidence intervals were considered significantly different. This approach was corroborated using standard chi-square tests in each time window (significant at $p < 0.05$).

**Theta sequence decoding analyses**. The objective of this approach was to determine whether the sequential activation of odor representations can be detected in a compressed form within a single theta cycle, and examine how this form of coding varies across trial types and across cycles. To do so, we used a multinomial logistic regression model, which while yielding weaker odor decoding than our CNN model (Supplementary Fig. 9b), can be used at the faster timescale necessary to decode within theta cycles (which is beyond the temporal resolution of the CNN model). Notably, the logistic regression model shows a similar pattern of sequential activation of upcoming stimuli within individual presentations as the CNN model, though, as expected, with more variability (Supplementary Fig. 10). Note that we used an unbiased approach and included the data from all active neurons (neurons were not filtered by their coding properties) and that a control analysis was performed to ensure group effects were not simply driven by particularly strong data from any one subject (see "Control analyses" below). Unless specified otherwise, an animal's ensemble decoding probabilities (the probability of A, B, C, and D) in bins of 10 or 120 degrees for each trial (i.e., each odor presentation) were treated as individual samples. Statistical analyses were performed on trial data pooled across subjects. To maximize statistical power, decoding

probabilities of past, present, and future stimuli were combined across odor B and C trials (i.e., odors ABC on B trials; odors BCD on C trials). A "within-ensemble" approach was used to examine how decoding varied across phases and stimulus types, which included both the use of paired t-tests (two-tailed) to probe specific a priori hypotheses and of repeated-measures ANOVAs to examine differences across levels (followed by posthoc tests controlling for the number of comparisons performed). Statistical significance was determined using $p < 0.05$.

*Logistic regression model*. We used a LASSO logistic regression model[56] that imposes $L1$ penalty on the regression parameters. For each trial (−2 s to 2 s, relative to port entry), the LFP signal from each tetrode was smoothed with a Butterworth filter between 4 Hz and 7 Hz (the main frequency range of theta we observed during odor sampling[26]) and the Hilbert transformation was then used to calculate the theta phase at each time point. The model was trained using the ensemble data from correct InSeq trials, specifically the firing rate of each neuron during the trough (120–240 degrees; CA1 pyramidal layer theta) of the first theta cycle beginning 100 ms after port entry (cycle 1). Trials in which the amplitude of that theta cycle was very weak (lowest 20th percentile) were not included in the analysis. Since not every neuron's activity was associated with odor presentations, the LASSO model automatically eliminated neurons not significantly contributing to the odor decoding (setting their beta weights to zero). The amount of penalty was set using a 10-fold cross-validation with stratification because the numbers of trials for different odors were unequal[57]. In the following analyses, the same model training was used but model testing was extended to other phases, cycles, and trial types.

*Past, present, and future decoding in cycle 1*. To apply the decoding model to the entire theta cycle, we used a sliding window (120 degrees in width) to calculate the firing rate of each neuron in an ensemble in 10-degree increments and determine the decoding probability of each odor per 10-degree bin. Note that this process was performed on each trial using the ensemble activity from the corresponding subject, and that trial data were then aggregated across subjects. Decoding traces were visualized in 10-degree bins but collapsed to 120-degree bins, or one value for each of the three phases, for hypothesis testing. Paired t-tests (two-tailed) were used to test the hypothesized pattern (Fig. 5a) of decoding for past, present, and future stimuli across the descending phase, trough, and ascending phase, respectively.

*Decoding across cycles*. Using the same method as above, we extended the decoding of the model to a series of theta cycles within each trial, ranging from three cycles before port entry to five cycles after (which consistently occurred toward the end of the odor presentation). For each InSeq trial (B and C trials combined), decoding probabilities of past, present, and future stimuli were obtained. Decoding probabilities for each stimulus type were then separately z-normalized (e.g., the past stimulus probability of a given trial was z-scored using all past stimulus probabilities across phases and cycles; see Fig. 6a). Note that this normalization was performed so plots could better capture their respective magnitude and dynamics across phases and cycles, but this transformation had no effect on the results: the same analyses were run using non-normalized values and the same pattern was obtained. For clarity, analyses focused on four cycles that were equidistant in time and spanned the duration of the trial (the second pre-stimulus cycle, cycle 1, cycle 3, and cycle 5). Data from those four cycles were collapsed across phase, aggregated across subjects, and analyzed using a two-way repeated-measures ANOVA (Stimulus × Phase), followed up by linear and quadratic trend analyses to further specify the dynamics across cycles (Fig. 6b).

*Trial-type comparisons*. To determine whether this form of theta sequence information is important for decision accuracy, decodings were compared between trials that were correctly or incorrectly identified by the animals. As in previous sections, this comparison focused on OutSeq trials (for better sampling of correct and incorrect trials) during the 250 ms time window preceding port entry (a period when behavior is matched between trial types), and focused on trials in the second or third sequence position. For each trial, the decoding probability of the expected (InSeq) stimulus across phases were calculated by averaging the values over two theta cycles occurring in that time window. Data from correct and incorrect trials were aggregated across subjects and analyzed using a two-way ANOVA (correct/incorrect × Phase; Phase as repeated factor) followed by posthoc unpaired t-tests at each phase (Bonferroni-corrected for three comparisons). Note that the result was the same when the comparison was downsampled (by repeating the analysis with 1000 permutations using 30 randomly-selected correct trials). To assess the flexibility of this form of theta sequences, we tested whether the same sequential information was decoded on InSeq and OutSeq trials: the same decoding pattern would indicate that these sequences rigidly reflect the most common sequence of items (InSeq stimuli) regardless of the stimulus presented on a given trial, whereas a different pattern would indicate decodings reflect the OutSeq nature of the individual trial. Using InSeq stimuli as the reference, we obtained the decoding probabilities across the three phases of cycle 1 for each InSeq and OutSeq trial (i.e., past InSeq stimulus in descending phase, present InSeq stimulus in trough, future InSeq stimulus in ascending phase). Data from InSeq and OutSeq trials (also from the second or third sequence position) were aggregated across subjects and analyzed using a two-way ANOVA (InSeq/OutSeq × Phase; Phase as repeated factor)

followed by posthoc unpaired *t*-tests at each phase (Bonferroni-corrected for three comparisons).

*Data visualization using single-cell decodings (cycle 1).* To help visualize the ensemble decoding results above, we used univariate logistic regression models for each neuron separately (Fig. 5b) with the same methods for model training and testing as above. Since the decoding of the present odor was stronger than past and future odors (as shown in Fig. 5a), we *z*-normalized the decoding probability of each stimulus separately to be able to reveal their respective distributions. More specifically, we used the distribution of past, present, and future stimulus decoding values across all phase bins and trials to *z*-score each decoding probability on a given bin and trial. We treated each individual neuron's model as a logistic classifier: for each 10-degree bin, we averaged the past, present, and future stimulus probabilities across trials and classified the stimulus with the highest *z*-score as the decoded stimulus. We then tested whether past, present, and future stimulus decodings followed the hypothesized pattern across phases. For clarity, we focused on neurons whose decodings showed significant modulation with phase for at least one odor type (62.2% of neurons) and then tabulated the proportion of phase-modulated neurons decoding the past, present, or future stimulus in each phase bin (grayscale plot in Fig. 5b). Similar to the temporal coding analyses, we then quantified the diagonal by computing the correlation between the columns (the three stimuli) and the rows (the three phases) and determined statistical significance using permutation testing (1000 random permutations). Specifically, we permuted the odor classifications within each phase bin, maintaining the overall distribution of decoded odors across the cycle but disturbing their phase. For each permutation, we computed the same correlation to build the null distribution of phase-permuted values.

*Theta phase precession in individual neurons.* We first examined theta phase precession using standard single-cell analyses quantifying how the phase of spikes shifted within odor presentations. More specifically, we matched a published approach[58] and tested whether each neuron showed a significant linear-circular correlation between spike time (from port entry to port withdrawal) and phase (0–360 degrees) using a significance threshold of *p* < 0.01 (see examples of statistically significant neurons in Supplementary Fig. 5). This analysis excluded neurons with fewer than 10 spikes during the trial period and was performed using data collapsed across odors (B–D) as well as using data from each odor separately (odor A trials were excluded to avoid the potentially confounding influence of running-associated theta). We then used a different approach to quantify how the decodings of individual neurons precessed within odor presentations. To do so, we adapted the single-cell decoding approach used in Fig. 5b and, for each animal separately, examined the decoding of the present stimulus across all neurons in 10-degree bins across cycles (ranging from three cycles before port entry to six cycles after). Specifically, for each bin, we computed for each neuron the mean present stimulus decoding across odor B and C trials, *z*-scored the decoding values using the mean and standard deviation of all neurons in the bin to put them on the same scale, and summed the *z* scores across neurons to capture the strength of the decoding in the population for each 10-degree bin. This process was then repeated for each cycle, which allowed us to obtain the phase at which present stimulus decoding peaked in each cycle. To facilitate comparisons across cycles, values were then normalized to the maximum decoding value of each cycle such that 1 and −1 indicated the strongest and weakest decoding of the present stimulus across neurons, respectively. We then aggregated the data across animals (5 data points per cycle) and determined if the observed peaks of present stimulus decoding varied in phase across successive cycles using a standard correlation analysis (Fig. 6d shows the data averaged across animals). We also examined this relationship using permutations, to maintain the overall strength of decodings during the trial period but disturb their phase. Specifically, to build the null distribution of correlation values, we permuted the order of theta cycles for each B or C trial, then recomputed the aggregate individual neuron decodings across theta cycles and correlation coefficient for each permutation (1000 random permutations).

### Control analyses

*Effect of excluding non-informative neurons on temporal coding results.* By default, our temporal coding analyses included all active neurons (neurons that fired at least one spike across any of the trial types). This unbiased approach allows us to be consistent across models: rather than selecting which cells go into the model (which could bias the results), we let the models determine which neurons are informative or uninformative. To ensure the pattern of results is the same using a less conservative approach, we reproduced the same analyses using only neurons with statistically significant temporal information. To do so, we quantified the temporal information provided by each neuron by running the model on each neuron independently. For each neuron, we then calculated the accuracy of reconstructed time and determined statistical significance by permuting the data (1,000 random permutations across the time factor). Using only the most informative neurons (the 81% that reached statistical significance; odor A: 144/173, odor B: 131/168, odor C: 142/175, odor D: 141/170), we re-ran the analyses previously performed on the entire ensemble of active neurons. We observed the same pattern of findings across analyses (compare Fig. 2 with Supplementary Fig. 6); in fact,

when non-significant cells were excluded, reconstructed time accuracy either remained unchanged (a difference < 2% for individual odors; Fig. 2b vs Supplementary Fig. 6b) or significantly increased (11.6% increase for the full sequence; Fig. 2e vs Supplementary Fig. 6e; $t_{(54)} = 2.836$, $p = 0.0064$).

*Effect of using a conservative 50:50 training/test validation on temporal coding results.* Neurons in PSTHs (Fig. 2a) are sorted using their average activity from the whole session. This is a standard approach in "time cell" studies[12,14] and an accurate parallel to our decoded time plots (Fig. 2b), which consider the whole session except for the trial being decoded (leave-one-out cross-validation). However, to confirm that significant temporal coding is observed using a more conservative training/test validation approach, we split the data into halves (three different types: first vs second half, Q1&Q3 vs Q2&Q4, and odd vs even trials). Using the data from one half, we sorted the PSTH and calculated temporal coding in the corresponding other half. As expected given the reduction in sample size, sorting and decoding using 50:50 splits led to noisier results, but time decoding accuracy was still significantly above permutation levels (Supplementary Fig. 7). Similarly, decoding accuracy for the full sequence was still significant using a 50:50 split when E was excluded to increase sampling (A–D: subjects' *p* values are 0.001, 0.042, 0.001, 0.001, and 0.001; whereas A–E primarily showed significant trends: subjects' *p* values are 0.01, 0.067, 0.1286, 0.062, and 0.046).

*Effect of subject on group decoding results.* We used the Cochran–Mantel–Haenszel test (a generalized version of the chi-square for three dimensions) to examine the effect of subject on group decoding accuracy. More specifically, we produced five group confusion matrices (each one leaving out a different subject), re-stratified the data by true odors, and tested whether predicted odors varied across the five leave-one-out matrices. We found that, given each odor, the accuracy of predictions did not significantly vary across group confusion matrices (CNN model: $\chi^2 = 3.7679$, df = 12, $p = 0.9873$; logistic regression model: $\chi^2 = 1.4718$, df = 12, $p = 0.9998$), confirming that the group effects reported are unlikely to be solely driven by any one subject.

*Effect of potential SWR-associated replay events on decoding.* Automated SWR (sharp-wave ripple) detection was performed by adapting a published approach[59]. A SWR event was identified as the co-occurrence of a sharp-wave event and a ripple event detected on two separate tetrodes, which were selected after visual inspection for maximal sharp wave (electrode closest to CA1 stratum radiatum) and ripple (electrode closest to the center of CA1 pyramidal layer) magnitude in their LFP. For both sharp wave and ripple events, the LFP was first filtered (sharp wave: high pass >4 Hz; ripple: bandpass 150–250 Hz) and instantaneous power was calculated as the real component of the Hilbert transformed trace. Putative events were identified as periods where power exceeded three standard deviations above the mean. Periods that occurred within 15 ms of each other were considered part of a common event. SWRs were then identified as periods of temporal overlap between putative sharp wave and ripple events, which were verified through visual inspection to remove artifacts. Temporal boundaries for SWR events were determined as the earliest and latest time points of either the sharp wave or ripple events identified. Using these criteria, only 18 trials (across all subjects) were identified as having a potential SWR event during the odor presentation (out of 1047 trials; < 2% of trials). The incidence was even lower in the 250 ms periods preceding and following the odor period (13 and 8 trials, respectively). The models were re-run with those trials excluded and the pattern of results did not change. Note that potential SWR events tended to cluster around the time of reward delivery (near the middle of the inter-odor interval), a period not included in our analyses.

**Reporting summary**. Further information on research design is available in the Nature Research Reporting Summary linked to this article.

## Data availability
The data used in this study have been deposited at https://doi.org/10.7280/D14X30[60]. Source data are provided with this paper.

## Code availability
All pre-processing and analyses were performed using Python 3.6, Matlab 2019b, and Prism 9. Code used for analysis and figure generation in this manuscript is available at https://doi.org/10.5281/zenodo.5579785[61]

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

## Acknowledgements

We thank members of our labs and departments for their helpful comments on this research. This work was supported by NIH (awards R01-MH115697, R01-DC017687, and T32-DC010775), NSF (awards CAREER IOS-1150292, DGE-1839285, and BCS-1439267), and the Whitehall Foundation (award 2010-05-84).

## Author contributions

N.J.F. conceived the project, conducted the experiment, funded the research, supervised the data analysis, and wrote the manuscript. B.S. funded the research, supervised data analysis, and wrote the manuscript. L.L., F.A., M.S., K.W.C., D.H., and G.A.E. performed

data curation and analysis, and reviewed and edited the manuscript. P.B. supervised the data analysis, and reviewed and edited the manuscript.

## Competing interests

The authors declare no competing interests.
