## [Peer Review File · Nature Communications]

Hippocampal ensembles represent sequential relationships among an extended sequence of nonspatial eventsREVIEWER COMMENTS

Reviewer #1 (Remarks to the Author):

The paper aims at understanding how the hippocampus codes neural activity in a non-spatial task. Similar to the same group's 2016 paper, the researchers taught the rat a sequence of odors and the rat had to learn whether an odor was part of the sequence, or rather there was a mismatch. While doing so, and in distinction from the previous paper, a large ensemble of neurons was recorded from the hippocampus, allowing for dimensionality reduction and classification techniques. The authors recorded many neurons in the task that responded to specific events within the task. Furthermore, the population activity resembled different aspects of the task at different time points: from stimulus identity, through existence of mismatch, to success in task, and this type of activity was predictive at the level of the ensemble. The neuron ensemble demonstrated predictive activity in response to odor presentation. Furthermore, anterograde and retrograde coding was apparent within the theta cycles.

While the paper contains some nice data and results, I have the following questions:

- 1) The paper does not track or look at the behavior of the Rat all. It is important to understand how the different results were contingent on the behavior, as when the rat stops holding the nose poke, place related activity may bulge in. Note that some of the results (for example comparing inSeq to outSeq trials) may be dependent on that, and thus it needs to be checked.
- 2) Note that the interpretation of temporal order (Fig. 3A) from comparison of in-seq and out-seq activity may be wrong, as it may be a mismatch or novelty signal and not a temporal-order signal. The two alternatives should be differentiated.
- 3) Temporal-information (similar to spatial information) should be measured for all cells, and significance of time cells should be determined using a shuffling procedure. Note that plots organized in a similar manner to Fig. 2A tend to have a strong diagonal (because of circularity) regardless of whether these are real "time cells" or not. One way to overcome this artefact of the eye in the display is to sort the cells using the first half of the recording and display the second half of the recording.
- 4) The description of the PSTH procedure on P. 4 line 22 is unclear.
- 5) P. 4 line 30 – "Importantly, reconstructed time accuracy was significantly higher before correct responses than incorrect responses" – was downsampling of correct responses performed to allow a fair comparison between correct and incorrect?
- 6) P. 5 line 18 – " the sequential organization of firing fields can be observed across the whole sequence" – is this not circular? Again, does it work when splitting the data in two (sorting according to one half of the experiment and displaying the second half)?
- 7) For the autoencoder used in Fig. 3 – why two dimensions were used? What happens if less or more dimensions were explored at the output?

8) Related to the comment on behavior above, an analysis should be done according to the withdrawal time of the rat in cases of a mismatch – how does this affect the representation?

9) P. 7 line 17 "the fourth window (750-1,000 ms) was when animals tended to receive feedback on whether their OutSeq decisions were correct" – what exactly is meant by "tended"?

10) Fig. 4 – it is advisable to show a scatter of a probability of each pair of predictions (i.e. C vs. D etc.) or a set of histograms for every odor/time combination. The random scatter is slightly confusing.

11) Did a variant of the logistic regression model, used for Fig. 5, work also on the Fig. 4 data?

12) P. 9 line 17 – " This analysis focused on the first theta cycle beginning 100 ms after odor onset" – What happened in other theta cycles?

Reviewer #2 (Remarks to the Author):

NCOMMS-20-21908-T

Hippocampal ensembles represent sequential relationships among an extended sequence of nonspatial events

Babak Shahbaba, Lingge Li, Forest Agostinelli, Mansi Saraf, Gabriel A. Elias, Pierre Baldi, and Norbert J. Fortin

Summary

Shahbaba et al. investigate the information reflected in hippocampal activity during an odor discrimination task composed of multiple instances of presentation of a sequence of odors interspersed with delays. Using extracellular recording in rats and several types of computational models, they show that hippocampal neurons represent time during the presentation of an odor, between presentations, and overall during trials; that these neurons also tend to be specific of a certain odor (conjunctive time/odor coding); that the representation of the next odor in a sequence can anticipate the actual presentation of the odor; that the quality of these representations is positively correlated with performance; and that the odor representations are organized similarly to spatial "theta sequences" with the previous odor represented more than the next odor at early phases of theta while this is reversed at late phases. This study very timely answers important questions about the extension of spatial findings to the non-spatial domain. It confirms recent findings that non-spatial parameters can be represented in the form of theta sequences, uses novel analytical tools that could be taken up by the

community to study neural representations, and in particular shows that the hippocampus can produce an anticipatory signal of future stimuli in the absence of specific sensory input, which if confirmed goes even beyond what had been shown of spatial theta sequences. It will certainly interest both spatial and non-spatial memory researchers as well as computational neuroscientists.

Strengths & Weaknesses

The manuscript is very well-written and easy to read, concise and logical; the task used (and previously published) is well-designed, the figures are very nicely presented and generally transmit a clear message. Some of the results are very interesting for example the fact that the quality of decoding is linked to behavioral performance (coherent with a recent preprint in the spatial field that the authors could cite (Zheng, Hwaun, and Colgin 2020)) and the anticipation of the next odor in the sequence (in spatial “theta sequences”, the animal can generally see future locations, while in this case the next odor is not directly perceptible).

On the other hand, the conciseness sometimes comes in the way of precision, and the reader is left with many methodological questions: what justifies the use of these different models, why not use the same one for all analyses? What is the analysis conducted on exactly, and what is the size of the sample? The similarities and differences between the present results and recently published ones (Allen et al. 2016; Terada et al. 2017) are not so clear (Terada et al., 2017 apparently found theta sequences of a non-spatial stimuli). As a disclaimer, I have no knowledge of convolutional neural networks, deep neural networks or multinomial logistic models so I cannot judge of their quality or reproducibility.

Detailed comments (major concerns)

Choice of model: A major concern is the justification behind the use of each analysis (modelling) approach used for answering different questions. It might look like the authors tried to answer a question with one model and when it did not work they switched to another type of model until it would give a significant result. The usual approach for answering these questions in the spatial literature would be Bayesian decoding, which the authors used for their first analysis. They could apply Bayesian decoding to answer each question, as a common thread throughout the manuscript, together with what they believe is the optimal approach for each question: this would 1) be more convincing and 2) provide a baseline to which compare the performance of these newer approaches. As a minimum, the authors should explain much more clearly the reasoning behind using each approach specifically for each question and why the other approaches would not be appropriate in each case (especially Bayesian decoding). Similarly, some analyses use only spikes (e.g. “latent representation analyses”) but others use spikes and LFP data (“neural decoding analyses”). Why is this the case?

Sample information & statistics: The manuscript is also missing clear information on how many cells were collected per recording session, how many trials were in each session, how many sessions did each rat perform, what are the criteria for including cells/trials/sessions in the analysis and, importantly, what sample is each analysis run on. Individual examples in the figures are given, which is good, but most of the time it is unclear what the error bar / confidence interval are computed over, and in some instances, it is not defined at all (e.g. SF2). It seems that many analyses are performed using trials as individual samples, while some of those trials will have been collected from the same rat and neurons and others will come from different rats and neurons. This mixes different amounts of within and between-individual variability and seems problematic as it does not show if the results are reproducible across sessions or individuals. If this was indeed the case, a single or a few sessions with many cells and a strong effect could be driving the effects observed. Instead I suggest that the authors do their analyses per experimental session and show example results (1 session, as seems to be already the case) as well as compute summary statistics (average results over all sessions) for each analysis. Of course, if the analyses are actually already performed on sessions, this should be clarified. A related concern is whether statistics were corrected for multiple comparisons as there is no mention of this anywhere in the manuscript.

Histology: I could not find any histology information, while given the implantation coordinates it is very possible that several of the cells recorded were from the CA2 field of hippocampus. Given that CA1 and CA2 have different properties (e.g. CA2 firing more during immobility (Kay et al. 2016)) it is important to estimate the origin of each neuron used in the analysis. If enough cells come from CA2 the authors should analyze those separately to assess if they have different properties from CA1.

Predictive coding: One of the most interesting and, to my knowledge, novel claims of the study is that the hippocampus can represent, during theta sequences, non-directly perceptible stimuli, in this case, anticipation of the next odor. The importance of this finding could be insisted upon in the manuscript; however, to make it really convincing the analysis would have to specifically show that this indeed happens before the next odor is presented, and in this task the delay between the current and next odor presentation varied so average graphs are not so informative. I recommend showing the time delay between start of presentation of the next odor and start as well as peak of the representation of a given odor to assess whether the representation indeed always starts before the presentation.

Theta sequences vs SWR-replay vocabulary: An important distinction to make throughout the manuscript is between Sharp-Wave/Ripple (SWR) – related “replay” and “theta sequences”. Both are believed to underlie different mechanisms and should not be mixed up. However, the term replay is sometimes used while the manuscript seems to focus on theta-related activity. Please replace it throughout the manuscript by sequence, reactivations, or an equivalent term; similarly, mentions of “theta-associated replay” should be replaced by “theta sequences” or perhaps “theta-associated representations” if the authors are not actually looking at the timescale of theta sequences - see for example the introduction of (Muessig et al. 2019).

Theta sequences vs SWR-replay analysis: Related to this, given that the authors do not specifically investigate sharp-wave/ripples related replay, but the animal is immobile and SWR are known to happen during immobility, they should either 1) detect and remove all SWR from all analyses so that it is clear that the results are not contaminated by SWR reactivations, or 2) extract those periods and analyze them separately to assess if and how their content differs from theta-related activity (note that if this was possible it would greatly increase the novelty factor of the study). As a minimum the authors should quantify the occurrence, frequency and location in task sequence of SWR or related reactivations so that the reader can assess whether those might be influencing the results or not. Unfortunately, according to the methods, signals were filtered between 154 Hz and 8.8kHz which would not be enough to detect ripples (their frequency can go up to 350Hz). The authors could instead detect hippocampal reactivations, in the form of a transient increase in multiunit activity - adapting for example the methods of (Carey, Tanaka, and van der Meer 2019) where a low theta power is used in conjunction with multiunit criteria - and remove them or analyze their content separately.

Detailed comments (minor concerns)

General

The current title is slightly unclear – what do the authors mean by “sequential relationships”? Does this mean continuity in time, order, contiguity in other parameters...? A more precise title would improve the understanding of the papers’ content, which is actually about conjunctive time by odor representations, theta sequences, and prediction of the future using some possibly novel computational approaches. The authors have not shown that the results generalize to other modalities than time.

Related to this, the vocabulary used is sometimes too vague, over-generalized (e.g. stimulus or event instead of odor, sequential relationships instead of order ...) or unclear (is trial sometimes used for condition? a trial generally means one instance of sampling but in fig S2 it seems that trial is used as condition “pairwise correlations between each odor A trial and each other A B C or D trial”). To clarify, the authors should use “odor” when they mean odor, and not stimulus; odor presentation for event, etc.. The authors should also give clear definitions of trial, sequence, session and related terms at the start of the result sections.

Please add a reference to the relevant methods section whenever a new method is used in the results.

It is slightly unclear if the data used here is new or reused from (Allen et al. 2016). I assume that this is new data but the authors should be more explicit about it (especially given that there are no author

contributions). This is important as the current study replicates some findings of the previous one such as time, odor and time by odor conjunctive coding, which should be emphasized in the discussion.

Most of the analyses rely on the behavior of the animals not varying throughout the different phases of the experiment. It is mentioned several times that “the location of the animal is constant” (e.g. p415, or “constant behavior” p716) but this is not shown anywhere. The authors should either show the stability of behavioral parameters that are known to influence hippocampal firing (position, speed, head-direction), or include those parameters in the analyses in order to control for them. Specifically, Methods state that one of the reward port was on the other side of the box, so the rat would have to move to go there.

It should be clarified if a unique sequence of odors had to be learned throughout the experiment (always A B C D E) or if this sometimes changes (regardless of the OutSeq trials) and if this specific sequence of odors was counterbalanced across rats. Also, if the same sequence is always used for each rat, they will associate an odor (A) to a position in the sequence (1st) so results showing odor-specific coding could actually show position-in-sequence coding. Did the authors disentangle these two interpretations, perhaps during the OutSeq trials?

Introduction

The authors should specify more clearly and without over-generalized language what is the purpose of their study and whether it aims to replicate past results (Allen et al. 2016; Terada et al. 2017) or bring in new information. P2 l20-22 states “it is critical to demonstrate that these coding properties: (i) extend to sequences of discrete events unfolding over several seconds, as in daily life episodes, and (ii) are linked to the successful retrieval of such event sequences”. Reactivation of extended sequences of events have been shown in the spatial domain by (Davidson, Kloosterman, and Wilson 2009) and a link between sequential theta representations and performance has recently been demonstrated in a preprint (Zheng, Hwaun, and Colgin 2020). Thus, the authors should acknowledge this and state their intent to extend these findings in the non-spatial domain. A list of the questions addressed, in order of their presentation in the results, at the end of introduction would be very useful, with the precise predictions tested.

P3l17 “at a higher level of abstraction” -> where is this demonstrated in the results?

Results

Fig 1. See my previous remark about behavioral information. This could be added here and in other figures.

“Neurons were sorted by their peak firing time in relation to the onset” should state “separately for principal neurons and interneurons”; also, why is the first interneuron inactive during A if this is the case, shouldn’t inactive cells be at the end, same for the first principal neuron? It would be good to show also in Fig 1 one example OutSeq sequence.

The figure should have a mention “adapted from (Allen et al. 2016)” as this seems to be the case.

P4L7-9: “we concentrated on the first four stimuli in the sequence to account for the decrease in sampling with sequence position produced by incorrect responses”. An illustration of the drop in sampling as a function of position in sequence would help support this claim.

P4I10 title: “CA1 ensemble activity is organized in sequences of firing fields within individual events, which carry event-specific information, and throughout event sequences” this is a bit obscure, please clarify.

P4I21 “peri-stimulus time histograms (PSTHs) involving the correct sorting were highly correlated across trials” – it is not clear what this means, but if it means that the activity of a given cell is highly correlated across several individual trials (a given presentation of odor), the authors should show individual trial data (spikes) to support this at least for some examples; if it means something else, please explain. Just assessing the correlation of sorted data is not convincingly showing time coding while demonstrating that these cells fire in that order across repeated trials is convincing.

P4I24-25 “all one-way ANOVAs and Dunnett’s [...]” As mentioned in main concerns, please specify what the tests are performed on here and later.

P4I30 “reconstructed time accuracy was significantly higher before correct responses than incorrect responses” it seems that in this case the model was trained in Inseq trials but Outseq trials were tested. This means that if the cells change their firing from Inseq to Outseq for some unrelated reason the decoder quality will drop when decoding Outseq. It seems that the model should be trained and tested on the same type of trials and compare correct vs incorrect in that category only. Also, this is an important result and a plot of these results should be shown.

Fig2. In “each peri-stimulus time histogram [...]” the authors could add that this is for each row of a matrix (or so it seems).

B,C,E: it would be good to show average chance level in C and the matrix obtained from random permutations in B and E, or at least 1 random permutation matrix obtained for 1 example session.

I suggest adding “time fields” or “time cells” in the figure title (or time by odor cells)

P5I30 “a trial corresponds to one odor presentation” As mentioned before, this clear definition of trial should appear at the start of results. Similarly, P6I21-23 “stimulus (which odor was presented), temporal order [...] identified the trial or not).”, these definitions could be moved (or repeated) at the start of results. Perhaps a different term than “temporal order” could be used for InSeq or OutSeq odor to distinguish this from the order of odor sequences, something like “order (mis)match”?

P6I23 “this approach simultaneously captured the temporal dynamics of each type of information differentiation within trials”: what does this mean? Could the authors be more specific?

P6I24 “As expected” – actually, one would probably expect no differentiation, but there is some differentiation so I would remove “as expected”.

P6I28 “income” should be “outcome”.

Fig3 Related to my previous comment about behavior, the percentage, or cumulated number of rats or sessions (it is unclear what the n is for this analysis but I assume it's session) doing action x or y as a function of time could be shown at the top below the “odor presentations” schematics, or any visualization of behavior that the authors deem appropriate.

In B, all plots should have the same y-axis limits to facilitate comparisons

P7I19: If rats received an auditory feedback about the outcome of their decision, how do we know that what is decoded is the outcome and not the perception of that signal? It seems that a trial-by-trial analysis time-locked on the outcome feedback could answer this question, is there a reason why the authors did not do this? This possible confound should be discussed in the discussion.

P7I21 “ensemble activity simultaneously differentiated information about [...]” – actually, it could be that separate neurons encode different types of information, and/or that LFP contains information about a specific parameter while neural data contains another. Unfortunately the approach used does not inform about what carries the information, only what information is there. Instead of the sentence above, the authors could say “population activity and LFP contains information about xx”. Related to this, if the authors could categorize cells that contribute the most to time coding, odor coding, time by

odor coding it would inform about the underlying mechanisms, while the present analysis only informs about what information is present in neural activity. Related to my comment about histology, it would be very interesting to see if different types of cells (time, odors) were in CA1 vs CA2.

P7125 “the sequential relationships among discontinuous events is replayed within individual events”
- “is” should be “are”, and as stated before “replay” should be changed into a different term.

P812 what is the 150-400ms window?

P8128-30 performance aspect: why is there no figure for this again? Also like mentioned previously stats should be done on sessions, not trials.

A Bayesian decoding approach of theta sequences would be more convincing, at least in parallel to the current approach which is a bit undirect (sequences could still happen but not necessary with the predicted relation to the theta cycle). Why did the authors not do this? I suggest at least showing some examples of single spike sequences.

Fig 5 “collapsed across rats” – again, does each rat only contribute 1 recording session?

How is the confidence interval computed?

It seems strange that there is no difference between previous or upcoming and current in any of the conditions in B, bottom - or were these just not tested?

P8125 it would be interesting to show data during presentation of A as a control (perhaps in supplementary).

Discussion

P1015 “novel” task: Given that this task has already been used and published several years ago (by them) the authors could choose another term.

P1018,18 “sequential firing fields (‘time cells’) : this formulation is a bit clumsy, the firing fields describe the activity pattern of the cell (generally from a rate map), but not the cell or type of cell, also many

other types of cells can have sequential activity, e.g. place fields. A more appropriate formulation would be “neurons expressing time-locked firing fields (‘time cells’) [...]”.

The similarities and differences between the current findings and past ones cited previously (see also (Kay et al. 2020 in the spatial domain)) should be discussed in more details here, especially whether non-spatial theta sequences were evidenced before, or anticipatory coding of a future, currently absent odor has previously been demonstrated. The author insist that a novel aspect of their task is that the sequence is discontinuous; if the rat does not change location and has to remain focused on the task during the whole presentation of the sequence, is there really a discontinuity? Arguments for the discontinuous aspect should be presented especially in relation to past experiments.

It seems that due to the training protocol rats are more familiar with the odors at the start of the sequence (given that the sequence stops when they make an error, they would have been exposed more to A B C than E, for example), and these odors seem to have a better decoding overall. This should be discussed (see SF2 and SF6 results).

Methods

General methods to add: What was the type of microdrive used? What tetrode wire & diameter? Were the electrodes plated? What is Neosporin for? What is the duration of the surgical recovery period?

P1817 “Institutional Animal Care and Use Committee”: could the authors indicate what country this applies to? Also, did the experiment not require ethical approval? Related note: in the editorial policy checklist, Additional policy considerations, because rats were involved in the study the 3rd checkbox should be “involved in the study”.

P18111-13 “Two water ports were used for reward delivery: one located under the odor port, the other at the opposite end of the track”: this is not referred to at all during the rest of the manuscript. What is the use of the second reward port, when did the rat go to it?

Supplementary

General: my previous remarks about stating the number of samples for each analysis, the nature of samples that the analyses are computed on, what do the error bar means etc. also apply to supplementary figures.

SF1: The three first diagonal plots look exactly identical, with the exception of some stretched / duplicated rows. This is very unexpected for biological data collected in different conditions, in this case different inter-trials. How do the authors explain this similarity?

SF2: How is the mean PSTH correlation computed? By PSTH do the authors mean matrix of PSTH? Are correlations computed between such matrices made from single trials? Please clarify or refer to the specific method section. The title “stimulus-specificity of sequential firing fields” could be more precise, like “evidence of time by odor coding”

SF5: what is T in the formula?

“The model was trained to minimize the difference between the original input data and the reconstructed data.” => is the model actually given the reconstructed data? In that case aren't the categories it finds constrained by what is chosen to be reconstructed? Perhaps the authors could explain more clearly what could be the pitfalls of this analysis and what controls they use to avoid those.

SF6: “used for the neural decoding analyses” -> Bayesian decoding was also used for some of the neural decoding analyses (time cells in Fig2) so this is a bit misleading

SF7: what does the “model training” overlay mean?

Review by Eleonore Duvelle.

References used:

Allen, Timothy A., Daniel M. Salz, Sam McKenzie, and Norbert J. Fortin. 2016. ‘Nonspatial Sequence Coding in CA1 Neurons’. *Journal of Neuroscience* 36 (5): 1547–63. <https://doi.org/10.1523/JNEUROSCI.2874-15.2016>.

Carey, Alyssa A., Youki Tanaka, and Matthijs A. A. van der Meer. 2019. ‘Reward Revaluation Biases Hippocampal Replay Content Away from the Preferred Outcome’. *Nature Neuroscience* 22 (9): 1450–59. <https://doi.org/10.1038/s41593-019-0464-6>.

Davidson, Thomas J., Fabian Kloosterman, and Matthew A. Wilson. 2009. ‘Hippocampal Replay of Extended Experience’. *Neuron* 63 (4): 497–507. <https://doi.org/10.1016/j.neuron.2009.07.027>.

Kay, Kenneth, Jason E. Chung, Marielena Sosa, Jonathan S. Schor, Mattias P. Karlsson, Margaret C. Larkin, Daniel F. Liu, and Loren M. Frank. 2020. ‘Constant Sub-Second Cycling between Representations

of Possible Futures in the Hippocampus'. *Cell* 180 (3): 552-567.e25.
<https://doi.org/10.1016/j.cell.2020.01.014>.

Kay, Kenneth, Marielena Sosa, Jason E. Chung, Mattias P. Karlsson, Margaret C. Larkin, and Loren M. Frank. 2016. 'A Hippocampal Network for Spatial Coding during Immobility and Sleep'. *Nature* 531 (7593): 185–90. <https://doi.org/10.1038/nature17144>.

Muessig, Laurenz, Michal Lasek, Isabella Varsavsky, Francesca Cacucci, and Thomas Joseph Wills. 2019. 'Coordinated Emergence of Hippocampal Replay and Theta Sequences during Post-Natal Development'. *Current Biology* 29 (5): 834-840.e4. <https://doi.org/10.1016/j.cub.2019.01.005>.

Terada, Satoshi, Yoshio Sakurai, Hiroyuki Nakahara, and Shigeyoshi Fujisawa. 2017. 'Temporal and Rate Coding for Discrete Event Sequences in the Hippocampus'. *Neuron* 94 (6): 1248-1262.e4.
<https://doi.org/10.1016/j.neuron.2017.05.024>.

Zheng, Chenguang, Ernie Hwaun, and Laura Lee Colgin. 2020. 'Impairments in Hippocampal Place Cell Sequences during Errors in Spatial Memory'. *BioRxiv*, April, 2020.04.20.051755.
<https://doi.org/10.1101/2020.04.20.051755>.

Reviewer #3 (Remarks to the Author):

This paper by Shahbaba et al. presents exciting findings about fine-scale temporal dynamics and hippocampal sequence coding beyond the spatial domain. It is especially interesting to see predictive information within the ensemble code, and refreshing that these data are obtained during a memory task with clear indicators of trial by trial performance. The authors are employing appropriately complicated analysis – and do a good job of explaining the details of the analysis and stepping through interpretations of the data. I have a few specific questions about the data that I hope may add clarity and sometimes augment the 'richness' of the manuscript – but I have no major criticisms of the main findings of the study.

1. When looking at the peri-stimulus histograms in Figure 2a, one thing that jumps out is that during the first 250 ms, neural activity does not seem odor specific. E.g. – for the top row, plots 2-4 (which are ordered by sequential activation aligned to A) for the neurons that fired in the first 250 ms when presented A, those same neurons fire at high rates during the first 250 ms of presentation of B,C,D – of course the sequence is perturbed. This seems to be true for all four rows. Is there a subset of cells that are simply indicating stimulus onset? This seems to keep coming up – i.e. in Figure 3a, we see this beautiful sphere for stimulus information. I wonder if the authors might consider commenting on this early on (if my interpretation is true).

2. Along these same lines, I am having a hard time reconciling these plots (in 2A) w/ the plot in 2D which shows the extended sequence across all odors. In this plot, I no longer see this propensity for some neurons to always fire within the first 250 ms of odor presentation. Is this because this is showing the 'median' onsets?

3. The analysis of data presented in Figure 3 are compelling. It's cool to see the cluster separation dynamics for different trial parameters – and especially neat to see some separation emerging early (like for Temporal Order and Trial Outcome). In the text, the authors point out that classification accuracy for Trial outcome and Temporal order are already significant between 250-500 ms. Would these be significant if a 99% cutoff were used? How do the authors interpret that Trial outcome classification falls again in the next temporal window despite improved Temporal order accuracy? Have the authors explored whether there are additional parameters represented in the hippocampal ensemble such as odor presentation and reward/no reward? I wonder if the increase seen in the Trial outcome in the last temporal window comes from conflation with reward signaling.

4. When seeing the data about temporal compression of odor sequence, I immediately wondered if neurons are showing theta phase precession during this task. Are they?

5. Is it possible to visualize an example of the theta sequences with raw data? I.e. replace Figure 5a w/ trial data showing a theta cycle w/ rasterized spike activity of neurons that had peak activation for the different odors?

6. Tying the theta sequence coding to the previous results – within a trial, the distinction of InSeq/OutSeq increased substantially between 500ms and 750ms – were theta sequences more robust during that time? (It is stated in the methods that this analysis was done on the first theta cycle beginning 100 ms after odor onset – so I guess I'm wondering if could explore theta sequences throughout the trial – which could potentially link theta sequences to decisions about whether the odors are in the proper sequence). Were theta sequences present during OutSeq trials? Were there differences in theta sequences between correct and incorrect trials?

Minor: Is odor onset and port entry the same? If so, please only use one in the main text.

Minor: Consider citing Arinov, ... Tank 2017, to reference other non-spatial sequence coding in hippocampus

NCOMMS-20-21908A: “Hippocampal ensembles represent sequential relationships among an extended sequence of nonspatial events” by Shahbaba, Li, Agostinelli, Saraf, Cooper, Haghverdian, Elias, Baldi, and Fortin.

Response to reviewers:

We are grateful for the reviewers' thoughtful and detailed comments. We deeply appreciate their enthusiasm about the novelty and significance of our findings, the state-of-the-art analytical approaches specifically designed to address each question, and the critical need to demonstrate that these fundamental coding properties extend to the nonspatial domain. We also thank them for suggesting additional analyses to help us further support our findings and conclusions, as well as enhance the richness of the manuscript (see updated Figs. 3 and 4, new Fig. 5 panels B-H, new Figs. S5-S7, and new Tables S1-S3). Finally, we appreciate the detailed comments aimed at further improving the clarity of the manuscript. Each comment is addressed in the point-by-point response below (in blue text), with the corresponding changes in the manuscript also identified in blue. We sincerely apologize for the delayed resubmission, which was due to a severe illness (the senior author underwent an extended course of cancer treatment).

Reviewer #1 (Remarks to the Author):

The paper aims at understanding how the hippocampus codes neural activity in a non-spatial task. Similar to the same group's 2016 paper, the researchers taught the rat a sequence of odors and the rat had to learn whether an odor was part of the sequence, or rather there was a mismatch. While doing so, and in distinction from the previous paper, a large ensemble of neurons was recorded from the hippocampus, allowing for dimensionality reduction and classification techniques. The authors recorded many neurons in the task that responded to specific events within the task. Furthermore, the population activity resembled different aspects of the task at different time points: from stimulus identity, through existence of mismatch, to success in task, and this type of activity was predictive at the level of the ensemble. The neuron ensemble demonstrated predictive activity in response to odor presentation. Furthermore, anterograde and retrograde coding was apparent within the theta cycles. While the paper contains some nice data and results, I have the following questions:

We thank the reviewer for the positive comments.

1) The paper does not track or look at the behavior of the Rat [at] all. It is important to understand how the different results were contingent on the behavior, as when the rat stops holding the nose poke, place related activity may bulge in. Note that some of the results (for example comparing inSeq to outSeq trials) may be dependent on that, and thus it needs to be checked.

The reviewer makes a good point that one analysis (the InSeq/OutSeq comparison in Fig. 3) could have been influenced by differences in behavior. On first submission, we could not think of a clear way to show the temporal dynamics across windows AND match the behavior on InSeq/OutSeq trials, so we focused on the former. We now have a solution: we added the same analysis but aligned to the port withdrawal response and revised Fig. 3 to show data aligned to both entry and withdrawal (focusing on four time windows in which behavior is matched across InSeq and OutSeq trials). The pattern of results did not change: the InSeq/OutSeq effect remained strong in the window preceding port withdrawal and the same temporal dynamics of information differentiation are still captured across time windows. The results (P7,L15 to P8,L18), methods (P29,L20), and Figure 3 have been updated accordingly. Thank you for leading us to a better solution.

Note that this potential concern does NOT extend to other analyses and figures as we carefully focused on time periods and trial types in which behavior was matched across conditions. For instance, the majority of analyses only included correct InSeq trials (so the behavior is identical across odor presentations; Figs. 2, 4, 5A,B,E-H) and, for analyses requiring other trial types, a specific window with matched behavior was targeted (e.g., 250ms preceding port entry for correct vs incorrect, first theta cycle of odor for InSeq vs OutSeq; Fig. 5C,D). This point is now emphasized on P4,L10.

2) Note that the interpretation of temporal order (Fig. 3A) from comparison of in-seq and out-seq activity may be wrong, as it may be a mismatch or novelty signal and not a temporal-order signal. The two alternatives should be differentiated.

In retrospect, we agree that we were not clear on this issue. The InSeq/OutSeq effect we showed is a match/mismatch signal *based* on temporal order information, but we had not shown that it *contained* temporal order information. We do so now by adding an analysis in which we show that, for both InSeq and OutSeq trials, the distance between clusters scales with the temporal order distance (lag) of trials. For instance, on InSeq trials, the distance between “B” and “C” clusters was smaller than between “B” and “D” clusters, and so on. On OutSeq trials, cluster distance increased with

the distance (in sequence position) between two OutSeq presentations of a given odor (e.g., D2 vs D3 < D2 vs D4). This result has been added to Fig. 3 (new panel B) and to the text (P8,L18). Notably, this finding is a nice parallel to the lag effect we showed in Fig. 2C using a different metric. We would not have thought of this without the reviewer's suggestion. Thanks!

3) Temporal-information (similar to spatial information) should be measured for all cells, and significance of time cells should be determined using a shuffling procedure. Note that plots organized in a similar manner to Fig. 2A tend to have a strong diagonal (because of circularity) regardless of whether these are real "time cells" or not. One way to overcome this artefact of the eye in the display is to sort the cells using the first half of the recording and display the second half of the recording.

We were concerned about the potential "circularity" problem as well, which is why we implemented the Bayesian algorithm and used a cross-validation approach (so the model is trained and tested using different trials). We agree that sorting any ensemble activity by peak time will give an impression of temporal structure (though not as strong as what we are showing); therefore, it is key to demonstrate (as we did) that an algorithm agnostic to this sorting can accurately reconstruct time from the ensemble activity alone. In other words, the only purpose of sorting the cells here is to help visualize the data -- the algorithm is not provided with that information.

We prefer to sort the cells using data from the whole session to match other "time cell" studies (e.g., MacDonald et al., 2011; Terada et al., 2017) and because it is a more accurate parallel to our reconstructed time plots (which consider the whole session except for the trial being decoded; leave-one-out cross-validation). We did, however, perform the requested analyses and report them in a new Figure S7. As expected given the reduction in sample size, sorting and decoding using 50:50 splits leads to noisier results, but time decoding is still significantly above permutation levels (added to text in "Control analyses" on P36,L1).

Finally, with regard to cell inclusion criteria, we used an unbiased approach similar to recent time cell papers (Terada et al., 2017; Mau et al., 2018). We prefer to include all simultaneously recorded cells and let the model determine which cells are informative or uninformative, rather than selecting which cells go into the model (which could bias the results). Importantly, this method also allows us to be consistent across models. Note that our approach is more conservative than the suggested alternative: in fact, we performed the requested analysis (determining significance of each cell using permutations) and show that time decoding accuracy either remained unchanged or slightly increased when non-significant cells were excluded (the new Fig. S6 reproduces Fig. 2 using only significant neurons). This is now mentioned in the Fig. 2 caption and in the text (see "Control analyses" on P35,L20).

4) The description of the PSTH procedure on P. 4 line 22 is unclear.

We clarified the sorting procedure for the PSTH in the text and in the methods (new section on PSTHs on P26,L10). The parallel with the detailed description in Fig. 2A should be clearer now.

5) P. 4 line 30 – "Importantly, reconstructed time accuracy was significantly higher before correct responses than incorrect responses" – was downsampling of correct responses performed to allow a fair comparison between correct and incorrect?

We used the Kolmogorov-Smirnov test for this comparison, which should be robust for smaller and/or unequal sample sizes (81 correct vs 33 incorrect trials). But to show that the results hold for a downsampled comparison, we repeated the analysis with 1,000 permutations using 33 randomly-selected correct trials. The result is still statistically significant: Kolmogorov-Smirnov $D = 0.364$, $P = 0.013$. This was added to the results (P5,L24) and methods (P28,L34). Note that we also added this downsampling approach to the other correct vs incorrect comparisons for the other analyses (P10,L17; P12,L11).

6) P. 5 line 18 – "the sequential organization of firing fields can be observed across the whole sequence" – is this not circular? Again, does it work when splitting the data in two (sorting according to one half of the experiment and displaying the second half)?

As mentioned in item #3 above, the key control for the potential "circularity" concern here is that the Bayesian algorithm (which is agnostic to the sorting of the cells) can significantly decode time within the whole sequence using the ensemble activity alone (Fig. 2E). Again, this decoding is done with cross-validation so the decoded trials are not used to train the model, further addressing this concern.

Because this particular analysis starts with only a subset of the data (only sequences in which all items were presented InSeq and in which all trials were correctly identified; ~14 sequences per animal), a 50:50 training/test validation is overly stringent. That is why we used a leave-one-out cross-validation, which is more appropriate in this case. Nonetheless, we quantified this further for the reviewer and found that the results were still significant when the 50:50

approach was used on a shortened sequence (A-D; excluding odor E provides an additional ~16 sequences per subject), whereas the results for A-E are significant until almost half the sequences were "left out" (6 out of 14). This was added to P36,L10. Related to point #3 above, time decoding in the full sequence improved when excluding non-informative cells (added to P35,L34).

7) For the autoencoder used in Fig. 3 – why two dimensions were used? What happens if less or more dimensions were explored at the output?

Reducing the data to two dimensions is the standard for this kind of approach because it facilitates data visualization. Further reduction would lead to substantial loss of information; adding dimensions would generally improve differentiation, but make it difficult to visually compare the patterns. We clarified this in the text (P29,L16).

8) Related to the comment on behavior above, an analysis should be done according to the withdrawal time of the rat in cases of a mismatch – how does this affect the representation?

Yes, thank you for the suggestion. This analysis is described in items #1 and #2 above.

9) P. 7 line 17 "the fourth window (750-1,000 ms) was when animals tended to receive feedback on whether their OutSeq decisions were correct" – what exactly is meant by "tended"?

The new analyses aligned to port withdrawal (Items #1, #2, and #8) eliminate this concern. Basically, because the original analysis was aligned only to port entry, most (but not all) OutSeq decisions were captured in that window.

10) Fig. 4 – it is advisable to show a scatter of a probability of each pair of predictions (i.e. C vs. D etc.) or a set of histograms for every odor/time combination. The random scatter is slightly confusing.

We apologize for the confusion --- we eliminated the random scatter by making the dots non-overlapping. The goal of this approach is to show the "raw data" (the decoding of each trial), with the counts in each colored dot cluster reflecting the decoding probabilities for each odor. We found histograms better suited to show central tendencies and dynamics (as we do in the bottom panel) than the raw data. We believe the plot is much more intuitive now. Again, thank you for leading us to a better solution.

11) Did a variant of the logistic regression model, used for Fig. 5, work also on the Fig. 4 data?

Yes. Although the logistic regression model has lower decoding accuracy relative to the CNN model, it showed a similar pattern during odor presentations (compare Fig. S10 with Fig. 4B, and Fig. 5F with Fig. 4D).

12) P. 9 line 17 – " This analysis focused on the first theta cycle beginning 100 ms after odor onset" – What happened in other theta cycles?

We considerably expanded analyses in that section of the manuscript (see the expanded Fig. 5). For clarity, these new analyses are described in the response to a similar comment from Reviewer 3 (see Rev3,#6).

Reviewer #2 (Remarks to the Author):

Summary

Shahbaba et al. investigate the information reflected in hippocampal activity during an odor discrimination task composed of multiple instances of presentation of a sequence of odors interspersed with delays. Using extracellular recording in rats and several types of computational models, they show that hippocampal neurons represent time during the presentation of an odor, between presentations, and overall during trials; that these neurons also tend to be specific of a certain odor (conjunctive time/odor coding); that the representation of the next odor in a sequence can anticipate the actual presentation of the odor; that the quality of these representations is positively correlated with performance; and that the odor representations are organized similarly to spatial “theta sequences” with the previous odor represented more than the next odor at early phases of theta while this is reversed at late phases. This study very timely answers important questions about the extension of spatial findings to the non-spatial domain. It confirms recent findings that non-spatial parameters can be represented in the form of theta sequences, uses novel analytical tools that could be taken up by the community to study neural representations, and in particular shows that the hippocampus can produce an anticipatory signal of future stimuli in the absence of specific sensory input, which if confirmed goes even beyond what had been shown of spatial theta sequences. It will certainly interest both spatial and non-spatial memory researchers as well as computational neuroscientists.

Strengths & Weaknesses

The manuscript is very well-written and easy to read, concise and logical; the task used (and previously published) is well-designed, the figures are very nicely presented and generally transmit a clear message. Some of the results are very interesting for example the fact that the quality of decoding is linked to behavioral performance (coherent with a recent preprint in the spatial field that the authors could cite (Zheng, Hwaun, and Colgin 2020)) and the anticipation of the next odor in the sequence (in spatial “theta sequences”, the animal can generally see future locations, while in this case the next odor is not directly perceptible).

On the other hand, the conciseness sometimes comes in the way of precision, and the reader is left with many methodological questions: what justifies the use of these different models, why not use the same one for all analyses? What is the analysis conducted on exactly, and what is the size of the sample? The similarities and differences between the present results and recently published ones (Allen et al. 2016; Terada et al. 2017) are not so clear (Terada et al., 2017 apparently found theta sequences of a non-spatial stimuli). As a disclaimer, I have no knowledge of convolutional neural networks, deep neural networks or multinomial logistic models so I cannot judge of their quality or reproducibility.

We thank the reviewer for the thoughtful and positive comments. We worked hard to make the paper clear and concise (despite the complexity of the work) and appreciate the reviewer’s help highlighting areas in which further details are needed. We addressed each point below.

Detailed comments (major concerns)

Choice of model: A major concern is the justification behind the use of each analysis (modelling) approach used for answering different questions. It might look like the authors tried to answer a question with one model and when it did not work they switched to another type of model until it would give a significant result. The usual approach for answering these questions in the spatial literature would be Bayesian decoding, which the authors used for their first analysis. They could apply Bayesian decoding to answer each question, as a common thread throughout the manuscript, together with what they believe is the optimal approach for each question: this would 1) be more convincing and 2) provide a baseline to which compare the performance of these newer approaches. As a minimum, the authors should explain much more clearly the reasoning behind using each approach specifically for each question and why the other approaches would not be appropriate in each case (especially Bayesian decoding). Similarly, some analyses use only spikes (e.g. “latent representation analyses”) but others use spikes and LFP data (“neural decoding analyses”). Why is this the case?

The choice of models was motivated by our scientific questions (this is why a strong collaboration between neuro and data scientists was critical to reveal the novel phenomena reported here). As detailed below, the same model could not be used to answer all questions -- we had to develop state-of-the-art models uniquely designed for each question -- though the results from the different models showed key points of convergence. We expanded the justification for each model (outlined below) in the corresponding sections of the results (P6,L24; P9,L8; P11,L3).

The first question was: can we quantify the degree of temporal information captured in the “time cell” coding we observed during the odor presentations? This was a simple question for which there exists a simple model (the Bayesian model used in Fig. 2; model #1), which is similar in logic to models used to quantify spatial information. However, that model architecture is not designed to answer more complex questions (like the ones discussed throughout the rest of the paper). To answer those questions, we had to develop more appropriate models.

The second question was: can we extract the different types of information simultaneously represented within odor presentations? We accomplished this by using an unsupervised model to reduce the dimensionality of the data to visualize (and analyze) the underlying patterns (model #2; Fig. 3). This is a powerful approach for differentiating patterns (as shown in Fig. 3), which is key here because we knew from prior work that odor selectivity tends to be more subtle and distributed in hippocampal ensembles than place information. Note that this is not something model #1 could accomplish since it is not designed for dimensionality reduction.

To follow that up, the third question was: can we identify which specific odor was represented at each moment in time? To answer this question, we used a convolutional neural network (CNN; model #3; Fig. 4) with a “supervised dimensionality reduction” approach (the model training includes information about which odor was presented on a given trial, which it uses to decode odor probabilities during testing). While this model is supervised (similar to model #1), it still uses dimensionality reduction (similar to model #2) to better capture and present the underlying patterns. This way, it combines the benefits of both models #1 and #2 to answer a more complex question. Notably, LFP information became relevant here because, unlike models #1 and #2, the architecture of model #3 could take advantage of multimodal input (spikes and LFP from the same tetrode).

The last question was: can we see these sequential representations in a temporally compressed form within individual theta cycles? This is not something model #3 could capture because it requires larger time windows than what is needed here. Instead, we used a simpler model (multinomial logistic regression) which has higher temporal resolution (at the cost of lower decoding accuracy). LFP information is also included here because of the need to link the decoding to the different phases of theta. Note that this type of model is more appropriate and common for the decoding of categorical variables (the different odors presented), whereas model #1 was designed for a continuous variable (time).

Finally, although the models have different architectures and were designed to answer different questions, there are notable points of convergence in the results of the different models (where one can see similar patterns across models), which we further highlighted in the revisions. For instance, models #1 and #2 separately captured the emergence of stimulus selectivity during trials (weak in 0-250ms window, strong in 250-500ms window; P5,L25 & P7,L26-32) and the temporal context (lag) effect (P5,L29 & P8,L18), whereas models #3 and #4 both captured the sequential reactivation of the series of nonspatial stimuli (P13,L3).

Sample information & statistics: The manuscript is also missing clear information on how many cells were collected per recording session, how many trials were in each session, how many sessions did each rat perform, what are the criteria for including cells/trials/sessions in the analysis and, importantly, what sample is each analysis run on. Individual examples in the figures are given, which is good, but most of the time it is unclear what the error bar / confidence interval are computed over, and in some instances, it is not defined at all (e.g. SF2). It seems that many analyses are performed using trials as individual samples, while some of those trials will have been collected from the same rat and neurons and others will come from different rats and neurons. This mixes different amounts of within and between-individual variability and seems problematic as it does not show if the results are reproducible across sessions or individuals. If this was indeed the case, a single or a few sessions with many cells and a strong effect could be driving the effects observed. Instead I suggest that the authors do their analyses per experimental session and show example results (1 session, as seems to be already the case) as well as compute summary statistics (average results over all sessions) for each analysis. Of course, if the analyses are actually already performed on sessions, this should be clarified. A related concern is whether statistics were corrected for multiple comparisons as there is no mention of this anywhere in the manuscript.

Thank you for bringing this to our attention. In retrospect, we agree that we had not provided a sufficient level of detail and clarity on this and the requested information was added as outlined below (organized by the reviewer’s key points). Our general approach (points 1-3 below) is now included in a new section called “Data inclusion, sampling, and pooling” (P25,L18) and in new Tables S1 and S2. We hope this clarifies our approach is rigorous and conservative.

1. We only used data from one session per animal to avoid oversampling neurons and trials. We are not combining an animal’s cells or trials across sessions to increase statistical power. The results thus reflect the activity of *simultaneously* recorded neural ensembles. This was added to the text (P25,L20) and to the captions of Tables S1-S2.

2. To balance the flexibility, power and rigor of the analyses, our statistical approach was built around using decoded probabilities from each trial as individual samples and performing comparisons using a “within-ensemble” design. For each analysis, we carefully considered *a priori* the most suitable unit of observation and approach to aggregate the data for group analyses. For the majority of the analyses, samples were individual trials (e.g., decoding probabilities for each odor on a given trial) and the trial data were then pooled across subjects. Importantly, comparisons were performed using a “within-ensemble” framework (e.g., how the pattern of decoding probabilities from each subject’s ensemble varied across time periods or trial types). This information was added to the new “Data inclusion, sampling, and pooling” section (P25,L22) and to the methods for each model (P27,L10; P29,L28; P30,L30; P32,L12). Note that this is a common approach because of its flexibility and robustness, but, as the reviewer noted, we should have clarified that the group effects are not simply driven by particularly strong data from any one subject. To do so, we added an analysis using the generalized Cochran-Mantel-Haenszel test (a version of the chi-square for three dimensions) which showed that removing data from any one subject did not significantly affect group decoding accuracy (see “Control analyses” on P36,L14).

3. We took a conservative approach to cell and trial inclusion criteria to avoid unintentionally biasing the analyses. As requested, we tabulated cell and trial counts per subject and clarified the inclusion criteria for each analysis (see the addition of Tables S1 and S2). Briefly, we only *excluded* cells if their firing rate values created problems with an analysis (e.g., zero spikes across all presentations of the odor of interest). Similarly, all trials were used except when there was a need to balance sample size across levels of a comparison (Fig. 2B) or meet a theta amplitude threshold to properly segment the activity in the different phases (Fig. 5). Note that filtering cells (or trials) based on their degree of informativeness would only *improve* the strength of our findings (e.g., response to Rev1,#3).

4. Other points. We have clarified the measures of uncertainty (confidence intervals or SEM), and how they were calculated, in each figure caption. We also clarified how standard approaches to correct for multiple comparisons fit into our statistical framework. Briefly, our approach focused primarily on using omnibus tests to examine all levels of a variable (e.g., repeated-measures ANOVAs) and on testing specific *a priori* predictions (e.g., Fig. 5A). We used standard correction procedures when ANOVA results were followed up by pairwise comparisons not defined *a priori* (e.g., Dunnett’s test in Fig. 2C, 3B; bonferroni-corrected *t*-tests in Fig. 5C,D). This information was added to the text (P25,L25) and in the methods for each model.

Histology: I could not find any histology information, while given the implantation coordinates it is very possible that several of the cells recorded were from the CA2 field of hippocampus. Given that CA1 and CA2 have different properties (e.g. CA2 firing more during immobility (Kay et al. 2016) it is important to estimate the origin of each neuron used in the analysis. If enough cells come from CA2 the authors should analyze those separately to assess if they have different properties from CA1.

We estimate that <10% of tetrodes could have been located in CA2 (histology for this dataset is included in Allen et al., 2016). Unfortunately, this is not enough CA2 data to properly examine how the coding properties of such ensembles differ from those in CA1. This information has been added to the “Data inclusion, sampling, and pooling” section (P26,L5). Note that this does not change the manuscript’s narrative, as we refrain from making direct comparisons across hippocampal subregions.

Predictive coding: One of the most interesting and, to my knowledge, novel claims of the study is that the hippocampus can represent, during theta sequences, non-directly perceptible stimuli, in this case, anticipation of the next odor. The importance of this finding could be insisted upon in the manuscript; however, to make it really convincing the analysis would have to specifically show that this indeed happens before the next odor is presented, and in this task the delay between the current and next odor presentation varied so average graphs are not so informative. I recommend showing the time delay between start of presentation of the next odor and start as well as peak of the representation of a given odor to assess whether the representation indeed always starts before the presentation.

New analyses with the logistic regression model now show this predictive coding in Fig. 5C, which is stronger on correct than incorrect trials in the ascending phase. However, it should be noted that this model has noisier decoding (rapid fluctuations) outside the odor period (which is to be expected given it is a different neural state than that used to train the model). The more powerful model (CNN; Fig. 4) is more robust against this and does show this form of predictive coding albeit at a lower temporal resolution (250ms bins). Producing the higher-resolution plot suggested would require a different experimental design. The fact that the predicted stimuli are not directly perceptible is indeed a novel claim of the study and this was added to P15,L22.

Theta sequences vs SWR-replay vocabulary: An important distinction to make throughout the manuscript is between Sharp-Wave/Ripple (SWR) – related “replay” and “theta sequences”. Both are believed to underlie different mechanisms and should not be mixed up. However, the term replay is sometimes used while the manuscript seems to

focus on theta-related activity. Please replace it throughout the manuscript by sequence, reactivations, or an equivalent term; similarly, mentions of “theta-associated replay” should be replaced by “theta sequences” or perhaps “theta-associated representations” if the authors are not actually looking at the timescale of theta sequences - see for example the introduction of (Muessig et al. 2019).

We agree that, to many, the term “replay” implies “SWR-associated replay,” which is why we used the term “theta-associated replay” (from Wickenheiser & Redish, 2014) to describe the general phenomenon we observed in Fig. 4 & 5. However, to avoid potential confusion, we now use the term “reactivation” instead of replay to refer to our findings (unless “theta-associated” is specifically mentioned). Note that the term “theta sequences” will mean something very specific to some readers (closer to what we show in Fig. 5 than in Fig. 4) so we prefer not to use it to refer to the general reactivation pattern (common to both Figs) as it may also lead to confusion.

Theta sequences vs SWR-replay analysis: Related to this, given that the authors do not specifically investigate sharp-wave/ripples related replay, but the animal is immobile and SWR are known to happen during immobility, they should either 1) detect and remove all SWR from all analyses so that it is clear that the results are not contaminated by SWR reactivations, or 2) extract those periods and analyze them separately to assess if and how their content differs from theta-related activity (note that if this was possible it would greatly increase the novelty factor of the study). As a minimum the authors should quantify the occurrence, frequency and location in task sequence of SWR or related reactivations so that the reader can assess whether those might be influencing the results or not. Unfortunately, according to the methods, signals were filtered between 154 Hz and 8.8kHz which would not be enough to detect ripples (their frequency can go up to 350Hz). The authors could instead detect hippocampal reactivations, in the form of a transient increase in multiunit activity - adapting for example the methods of (Carey, Tanaka, and van der Meer 2019) where a low theta power is used in conjunction with multiunit criteria - and remove them or analyze their content separately.

Good point. We spent considerable time implementing and validating SWR-detection algorithms to quantify their distribution during task performance (note the LFP filtering went up to 8.8 kHz, not 8.8 Hz, so enough to detect SWRs). We found that trials with potential SWR events during odor presentations were very rare: 18 trials across all subjects (out of 1,047 trials; <2% of trials). The incidence was even lower in the 250ms periods preceding and following the odor period (13 and 8 trials, respectively). Removing the few affected trials from each analysis did not change the results (see “Control analyses” on P36,L22). Note that, as expected, SWR events tended to cluster around the time of reward delivery (near the middle of the inter-odor interval); however, decoding the content of SWR events will have to be done in a separate study, as our current models are not designed to handle that level of temporal resolution.

Detailed comments (minor concerns)

General

The current title is slightly unclear – what do the authors mean by “sequential relationships”? Does this mean continuity in time, order, contiguity in other parameters...? A more precise title would improve the understanding of the papers’ content, which is actually about conjunctive time by odor representations, theta sequences, and prediction of the future using some possibly novel computational approaches. The authors have not shown that the results generalize to other modalities than time.

Just like humans do, our rats have learned the sequential relationships among a series of nonspatial events that are separated in time. They learned that A leads to B, that B leads to C, and so on. The novel forms of coding reported here directly support this capacity: the “conjunctive time by odor representations” (specifically the lag effect in Fig. 2C that varied with the sequence position of the odors), the temporal coding across the whole sequence (Fig. 2D,E), the “prediction of the future” (the next item in the sequence), and the “theta sequences” (the compression of those sequential relationships within a theta cycle). It has been hypothesized for a long time that these neural mechanisms may support this capacity, but this study is the first to directly show it. Nevertheless, we took another look at the title but could not come up with a better way to capture this (we are open to suggestions). Finally, we are unsure how to show that our results extend to other modalities than time: sequences are inherently temporal and a unique aspect of our nonspatial approach is that we are isolating time from space.

Related to this, the vocabulary used is sometimes too vague, over-generalized (e.g. stimulus or event instead of odor, sequential relationships instead of order ...) or unclear (is trial sometimes used for condition? a trial generally means one instance of sampling but in Fig. S2 it seems that trial is used as condition “pairwise correlations between each odor A trial and each other A B C or D trial”). To clarify, the authors should use “odor” when they mean odor, and not stimulus; odor presentation for event, etc.. The authors should also give clear definitions of trial, sequence, session and related terms at the start of the result sections.

Our general approach is to use specific terms (e.g., odor) when describing methods, but broader terms (e.g., stimulus) when referring to general principles or implications of the findings that extend beyond our specific methods. The “trial”, “sequence”, and “session” terminology is now emphasized earlier in the manuscript (first paragraph of Results) and clarified at a few additional key locations in the manuscript (in addition to the “Data inclusion, sampling and pooling” section). We have modified the Figure S2 caption to resolve the potential confusion between trials and conditions.

Please add a reference to the relevant methods section whenever a new method is used in the results.

References added.

It is slightly unclear if the data used here is new or reused from (Allen et al. 2016). I assume that this is new data but the authors should be more explicit about it (especially given that there are no author contributions). This is important as the current study replicates some findings of the previous one such as time, odor and time by odor conjunctive coding, which should be emphasized in the discussion.

It is the same raw data as Allen et al., (2016; stated in first paragraph of results) but that first paper only scratched the surface of this complex dataset. Basically, it focused on introducing this novel dataset and characterizing the InSeq vs OutSeq contrast using traditional single-cell analyses (and some LFP). There was no decoding of the information represented in the ensemble activity or of the dynamics of that information within trials (as that required the extensive model development we did here) and thus no overlap with the present results. To clarify, there was no time coding in that paper either -- what the reviewer is referring to are cells that responded to the InSeq/OutSeq contrast in an item-specific way, which is different from the time (temporal) coding reported here. As requested, we clarified how the present results extended those earlier findings in the discussion (P14,L27).

Most of the analyses rely on the behavior of the animals not varying throughout the different phases of the experiment. It is mentioned several times that “the location of the animal is constant” (e.g. p415, or “constant behavior” p716) but this is not shown anywhere. The authors should either show the stability of behavioral parameters that are known to influence hippocampal firing (position, speed, head-direction), or include those parameters in the analyses in order to control for them. Specifically, Methods state that one of the reward port was on the other side of the box, so the rat would have to move to go there.

The analyses reported here focus on when the animal's nose is in the odor port. Note that only one odor port is used for all odor presentations and that it is just large enough to fit their nose (so the location of the animal is consistent across trial types). Yes, there are moments when the animal runs around (they are required to go to the reward port in the back between sequence presentations), but those data are not included here. The only analysis in which behavior differed between trial types was the InSeq vs OutSeq comparison in the original Figure 3 (now corrected; see Rev1,#1).

It should be clarified if a unique sequence of odors had to be learned throughout the experiment (always A B C D E) or if this sometimes changes (regardless of the OutSeq trials) and if this specific sequence of odors was counterbalanced across rats.

Only one sequence was used during training (ABCDE; same odors and order across rats) and the same sequence was used in the recorded session. Although we have counterbalanced sequences in behavioral studies using the same task (e.g., Allen et al., 2014, 2015), we chose to match the sequences across animals in this recording study to keep the neural coding as consistent as possible across rats. This information was added to the “Behavior” section on P24,L6.

Also, if the same sequence is always used for each rat, they will associate an odor (A) to a position in the sequence (1st) so results showing odor-specific coding could actually show position-in-sequence coding. Did the authors disentangle these two interpretations, perhaps during the OutSeq trials?

Although it is possible that information about the associated sequence position contributes to the odor-specific coding we observed, two of our results suggest the odor is the key information. First, as mentioned earlier (Rev3,#1), trial-type differentiation sharply increases around 250ms (when information about the delivered odor is expected to reach the hippocampus); in contrast, information about sequence position is present from the beginning of the trial and yet there is little differentiation at that moment (see Fig. 2A and 3). Second, we observed a significant difference in the decoding of InSeq and OutSeq trials (Fig. 5D). Since this analysis matched sequence position, this difference reflects the fact that distinct odors are being presented. These two points were added to the results (P8,L2 and P12,L17).

Introduction

The authors should specify more clearly and without over-generalized language what is the purpose of their study and whether it aims to replicate past results (Allen et al. 2016; Terada et al. 2017) or bring in new information. P2 I20-22 states “it is critical to demonstrate that these coding properties: (i) extend to sequences of discrete events unfolding over several seconds, as in daily life episodes, and (ii) are linked to the successful retrieval of such event sequences”. Reactivation of extended sequences of events have been shown in the spatial domain by (Davidson, Kloosterman, and Wilson 2009) and a link between sequential theta representations and performance has recently been demonstrated in a preprint (Zheng, Hwaun, and Colgin 2020). Thus, the authors should acknowledge this and state their intent to extend these findings in the non-spatial domain. A list of the questions addressed, in order of their presentation in the results, at the end of introduction would be very useful, with the precise predictions tested.

To clarify, all presented findings are novel and represent significant advances. We provided points of convergence with past results but those are not simply replications: we are revealing how previously reported coding properties extend to completely different dimensions or cognitive demands. We suspect the lack of clarity here is due to the fact that we did not directly state that we are not only extending previous spatial work (as the reviewer noted) but previous nonspatial work as well. We added a few points of clarification in the introduction to further highlight the rationale and significance of this study. We also added the Davidson et al. and Zheng et al. citations (the Allen et al. and Terada et al. citations were already included).

In terms of listing our series of specific questions in the introduction, we are concerned this approach is not well suited to this particular manuscript because of the complexity of the experiment and analyses. We are currently stating general questions in the introduction, but the specific questions (and associated predictions) come in the results when the reader has received the necessary information to put the question into context.

P3I17 “at a higher level of abstraction” -> where is this demonstrated in the results?

It was referring to the subsequent clause “including temporally compressed representations of task-critical sequential relationships” which is what we show in Fig. 4 & 5. We changed the sentence (now P3,L20) to clarify what is being abstracted.

Results

Fig. 1. See my previous remark about behavioral information. This could be added here and in other figures. “Neurons were sorted by their peak firing time in relation to the onset” should state “separately for principal neurons and interneurons”; also, why is the first interneuron inactive during A if this is the case, shouldn't inactive cells be at the end, same for the first principal neuron? It would be good to show also in Fig. 1 one example OutSeq sequence. The figure should have a mention “adapted from (Allen et al. 2016)” as this seems to be the case.

We were not sure where to put the first pyramidal neuron and interneuron because they did not fire during odor A, so we put them at the top. In hindsight, it makes more sense to have them at the bottom, as suggested, so we made the change to the figure and caption. We also added that principal neurons and interneurons were sorted separately.

We thought about adding an example of representative activity for an OutSeq sequence, but feel it would unnecessarily complicate the figure. InSeq activity is the primary focus of the paper, whereas OutSeq activity is only relevant for a subset of analyses (and an OutSeq example is unlikely to enhance the clarity of these analyses).

We will check with the copy editor about whether this figure should mention “adapted from Allen et al., 2016”. This figure was created specifically for this paper, though we maintained some of the generic elements of the drawings from previous papers for consistency (e.g., color scheme, rat drawing, odor delivery). In that sense, similarities with figures from our previous papers are only superficial.

P4L7-9: “we concentrated on the first four stimuli in the sequence to account for the decrease in sampling with sequence position produced by incorrect responses”. An illustration of the drop in sampling as a function of position in sequence would help support this claim.

Thank you for bringing this up. Trial counts are now shown in Table S1 (there were 42 fewer trials of E than of D). But we should note that this was not an arbitrary decision -- it was corroborated by an examination of the confusion matrices for the CNN and logistic regression models (which show how well the models correctly identified the different odor trials). Essentially, we noted that the decoding of Odor E was considerably lower (0.15 or below) than that of the odors (0.32 or above; compare with diagonal in Fig. S9B). Excluding Odor E therefore allowed us to focus on the odors

with the most balanced decoding across analyses. This was added to Table S1 where information about trial counts is provided.

P4I10 title: “CA1 ensemble activity is organized in sequences of firing fields within individual events, which carry event-specific information, and throughout event sequences” this is a bit obscure, please clarify.

We clarified the heading. Thank you.

P4I21 “peri-stimulus time histograms (PSTHs) involving the correct sorting were highly correlated across trials” – it is not clear what this means, but if it means that the activity of a given cell is highly correlated across several individual trials (a given presentation of odor), the authors should show individual trial data (spikes) to support this at least for some examples; if it means something else, please explain. Just assessing the correlation of sorted data is not convincingly showing time coding while demonstrating that these cells fire in that order across repeated trials is convincing.

The confusion likely arises from an error in that sentence (we said comparisons were performed across columns, but it was across rows), which we corrected. Thank you for catching this error. Essentially, we showed that PSTHs from individual trials (the neuron by time bin matrix from each odor presentation) were more highly correlated across trials of the same odor type than trials of different odor types. More specifically, this analysis was done one “row” at a time (referring to Fig. 2A): comparison of odor A trials involved A vs A, A vs B, A vs C, A vs D and kept the same cell sorting (to the onset of odor A in this case). This has been clarified in the results (P5,L6), methods (new section on P26) and in the caption of Fig. S2A. This (model-independent) analysis, combined with the corresponding analysis using the Bayesian model in the next sentence, provide strong evidence that this temporal organization is stimulus-specific.

P4I24-25 “all one-way ANOVAs and Dunnett’s [...]” As mentioned in main concerns, please specify what the tests are performed on here and later.

Expanding on the answer above, the one-way ANOVAs are performed on the r values obtained when comparing each pair of trial-specific PSTHs (e.g., each A trial vs each A trial, each A trial vs each B trial, each A trial vs each C trial, each A trials vs each D trial) while keeping the neuron sorting constant (in this example, to the onset of odor A). Fig. S2A shows the mean r values for each type of pairwise comparison. Each ANOVA was statistically significant and was followed up by a Dunnett’s test to confirm the effect was driven by “same” trials being more strongly correlated than “other” trials. This information was added to the methods (new paragraph on correlation analyses on P26,L22) and in the Fig. S2 caption.

P4I30 “reconstructed time accuracy was significantly higher before correct responses than incorrect responses” it seems that in this case the model was trained in Inseq trials but Outseq trials were tested. This means that if the cells change their firing from Inseq to Outseq for some unrelated reason the decoder quality will drop when decoding Outseq. It seems that the model should be trained and tested on the same type of trials and compare correct vs incorrect in that category only. Also, this is an important result and a plot of these results should be shown.

We know from our previous behavioral paper (Allen et al., 2014) that it is best to use OutSeq trials for correct vs incorrect comparisons. In fact, InSeq mistakes are rare and tend to occur few ms before the signal, thus most likely reflecting errors of anticipation instead of incorrect decisions (added to methods on P28,L21). Because decoding takes place before the odor is presented (i.e., 250ms before port entry), we are avoiding potential concerns about the effect being driven by the OutSeq activity or decoding across trial types (as there is no difference between InSeq and OutSeq trials at that time point; added to results on P5,L23).

Since this result is a simple bar plot (see right; lower values indicate greater accuracy) we feel that mentioning it in the text is sufficient (adding it to Fig. 2 makes the bottom row too small)..

Fig2. In “each peri-stimulus time histogram [...]” the authors could add that this is for each row of a matrix (or so it seems).

Yes, as noted above (point P4I21), this has been clarified in the text. We also clarified it in the caption.

B,C,E: it would be good to show average chance level in C and the matrix obtained from random permutations in B and E, or at least 1 random permutation matrix obtained for 1 example session.

We added a chance level (random permutations) in Fig. 2C and Fig. S2B. Permuted matrices are important for the statistics but do not tend to be visually informative (see example on the right for data in Fig. 2E; 1,000 random permutations) and are thus not included in the paper.

I suggest adding “time fields” or “time cells” in the figure title (or time by odor cells)

“Time fields” was added to the figure title.

P5I30 “a trial corresponds to one odor presentation” As mentioned before, this clear definition of trial should appear at the start of results. Similarly, P6I21-23 “stimulus (which odor was presented), temporal order [...] identified the trial or not).”, these definitions could be moved (or repeated) at the start of results. Perhaps a different term than “temporal order” could be used for InSeq or OutSeq odor to distinguish this from the order of odor sequences, something like “order (mis)match”?

The definition of a trial was clarified in the first paragraph of the results (P4). The addition of the lag analysis (Fig. 3B) should make the term “temporal order” more accurate and informative.

P6I23 “this approach simultaneously captured the temporal dynamics of each type of information differentiation within trials”: what does this mean? Could the authors be more specific?

It is referring to how the different types of information mentioned in the previous sentence (stimulus, temporal order, and trial outcome) varied within a trial. The next sentence then starts describing the specific patterns. We removed the word “differentiation” to simplify the sentence.

P6I24 “As expected” – actually, one would probably expect no differentiation, but there is some differentiation so I would remove “as expected”.

Thanks, we made the correction.

P6I28 “income” should be “outcome”.

Thanks, we made the correction.

Fig3 Related to my previous comment about behavior, the percentage, or cumulated number of rats or sessions (it is unclear what the n is for this analysis but I assume it's session) doing action x or y as a function of time could be shown at the top below the “odor presentations” schematics, or any visualization of behavior that the authors deem appropriate.

This figure has been revised (see Rev1,#1) and information about how the data is aggregated has been added to the caption and methods.

In B [now C], all plots should have the same y-axis limits to facilitate comparisons

The objective of this analysis is to capture the range (from chance to peak levels) of classification accuracy and compare the different dynamics across plots. For that reason, we feel it is more appropriate to keep the scaling as is. Having the plots on the same y-limits would add a lot of white space because of different chance and peak levels.

P7I19: If rats received an auditory feedback about the outcome of their decision, how do we know that what is decoded is the outcome and not the perception of that signal? It seems that a trial-by-trial analysis time-locked on the outcome feedback could answer this question, is there a reason why the authors did not do this? This possible confound should be discussed in the discussion.

This is related to Rev3,#3. The auditory feedback is only available after the animal has made its response so it cannot account for the significant trial outcome differentiation we show during the odor presentation (which is a key finding here). We did not mean to imply that we could dissociate trial outcome and auditory feedback after the response (we cannot); in fact, we argue that the auditory feedback likely *drives* the peak in trial outcome differentiation observed after trials, as it is an external signal confirming the trial outcome. We clarified this in the text (P8,L15).

P7I21 “ensemble activity simultaneously differentiated information about [...]” – actually, it could be that separate

neurons encode different types of information, and/or that LFP contains information about a specific parameter while neural data contains another. Unfortunately the approach used does not inform about what carries the information, only what information is there. Instead of the sentence above, the authors could say “population activity and LFP contains information about xx”.

The model used in that section of the results does NOT include LFP activity, so we kept the original sentence.

Related to this, if the authors could categorize cells that contribute the most to time coding, odor coding, time by odor coding it would inform about the underlying mechanisms, while the present analysis only informs about what information is present in neural activity. Related to my comment about histology, it would be very interesting to see if different types of cells (time, odors) were in CA1 vs CA2.

We show the “raw data” (Fig. 2) so the reader can get a sense of the proportion of the ensemble exhibiting time, odor, and time-in-odor coding and their dynamics. However, we refrain from categorizing/labeling each cell because doing so would require the use of arbitrary cut-offs. Not only is this tricky given the graded nature of odor coding, the actual proportions of cells would vary substantially depending on the cut-off used. In short, categorizing individual cells is a good idea in theory (and we thought about implementing this originally), but, in the end, we felt it did not provide additional mechanistic information than what is already provided in the raw data we show (the key quantification is the amount of information contained in the ensembles). The point about comparing CA1 and CA2 has been addressed above.

P7125 “the sequential relationships among discontinuous events is replayed within individual events” - “is” should be “are”, and as stated before “replay” should be changed into a different term.

The change has been made.

P812 what is the 150-400ms window?

It is the time window from which neural activity was used to train the CNN model, which was defined *a priori* as the period most likely isolating the processing of the currently presented odor.

P8128-30 performance aspect: why is there no figure for this again? Also like mentioned previously stats should be done on sessions, not trials.

As in the comment above (P4130), this is a simple bar plot (see right; high values indicating greater accuracy) that would disrupt the layout of the figure (Fig. 4). We feel that mentioning it in the text is sufficient because the bar plot adds little information to the statistical result we report. As mentioned earlier, we only used data from one session per animal (to avoid oversampling) so the effect is examined by comparing correct and incorrect trials.

A Bayesian decoding approach of theta sequences would be more convincing, at least in parallel to the current approach which is a bit indirect (sequences could still happen but not necessary with the predicted relation to the theta cycle). Why did the authors not do this? I suggest at least showing some examples of single spike sequences.

As mentioned above (in Choice of model), the Bayesian algorithm (the one used in Fig. 2) is designed for a continuous variable (i.e., time). Decoding categorical variables like odors is best accomplished using a different model (the multinomial logistic regression model used in Fig. 5), which we optimized for this specific problem (i.e., it provides the most “direct” approach). Decoding examples have been added to Figure 5B (see response to Rev3,#5).

Fig. 5 “collapsed across rats” – again, does each rat only contribute 1 recording session? How is the confidence interval computed?

These points have been addressed above (see “#4. Other points”).

It seems strange that there is no difference between previous or upcoming and current in any of the conditions in B, bottom - or were these just not tested?

The key aspect of the prediction here (the one defined *a priori*) is the relationship between the decoding of past vs future stimuli in the descending and ascending phases. Posthoc comparisons with the current stimulus decoding in those phases are not central to the hypothesis and therefore not tested.

Discussion

P10I5 “novel” task: Given that this task has already been used and published several years ago (by them) the authors could choose another term.

Good point. This has been fixed.

P10I8,18 “sequential firing fields (‘time cells’) : this formulation is a bit clumsy, the firing fields describe the activity pattern of the cell (generally from a rate map), but not the cell or type of cell, also many other types of cells can have sequential activity, e.g. place fields. A more appropriate formulation would be “neurons expressing time-locked firing fields (‘time cells’) [...]”.

The change has been made.

The similarities and differences between the current findings and past ones cited previously (see also (Kay et al. 2020 in the spatial domain)) should be discussed in more details here, especially whether non-spatial theta sequences were evidenced before, or anticipatory coding of a future, currently absent odor has previously been demonstrated. The author insist that a novel aspect of their task is that the sequence is discontinuous; if the rat does not change location and has to remain focused on the task during the whole presentation of the sequence, is there really a discontinuity? Arguments for the discontinuous aspect should be presented especially in relation to past experiments.

A new paragraph has been added to the discussion (P14,L23) to provide a more detailed description of how the present results extend past ones (the Kay et al., 2020 reference was also incorporated). The form of nonspatial theta sequence we report has not been previously demonstrated (P15,L30). Terada et al. (2017) showed theta sequences that represented immediately preceding and upcoming time points (on the order of a few hundred ms), whereas we show a higher level of abstraction: that the sequential relationships among past, present, and future events (separated by seconds in real time) can also be compressed within a theta cycle. The sequential reactivation of the predicted sequence of nonspatial stimuli has not been previously reported; it has only been shown with spatial information (P15,L21).

A key feature of the task is that it involves sequences of stimuli that are discontinuous (with a “g”, which is different from discontinuous) in that they are separated in time (by a few seconds) and not directly perceptible (as the reviewer pointed out), which parallels cognitive demands in humans. We do not claim that their experience of the sequence is discontinuous, although we did incorrectly use that word once which may have led to this confusion (error was corrected).

It seems that due to the training protocol rats are more familiar with the odors at the start of the sequence (given that the sequence stops when they make an error, they would have been exposed more to A B C than E, for example), and these odors seem to have a better decoding overall. This should be discussed (see SF2 and SF6 results).

Good point. It is true that, given the stepwise training, animals have had more exposure to the earlier odors in the sequence in their lifetime. However, all odors are highly familiar to them (they have received hundreds of presentations of each during training), so we view differences in decoding levels as having more to do with sampling idiosyncrasies of the recorded session (the number of trials for each odor and the coding properties of those particular ensembles). For instance, decoding of odor D is slightly higher than decoding of odor C (Fig. S9B), whereas their training history would predict the opposite. Note that we included this information with other important characteristics of the behavior in the “Behavior” section of the methods (P24,L9), instead of making it an isolated statement in the discussion.

Methods

General methods to add: What was the type of microdrive used? What tetrode wire & diameter? Were the electrodes plated? What is Neosporin for? What is the duration of the surgical recovery period?

The details have been added to the methods (P25,L2 and P24,L34).

P18I7 “Institutional Animal Care and Use Committee”: could the authors indicate what country this applies to? Also, did the experiment not require ethical approval? Related note: in the editorial policy checklist, Additional policy considerations, because rats were involved in the study the 3rd checkbox should be “involved in the study”.

The institutions for IACUC approval were added (Boston University and University of California, Irvine; P23,L8). The boxes of the Editorial Policy Checklist mentioned were checked. Thank you for catching that.

P18I11-13 “Two water ports were used for reward delivery: one located under the odor port, the other at the opposite

end of the track”: this is not referred to at all during the rest of the manuscript. What is the use of the second reward port, when did the rat go to it?

The second reward port is not relevant to the analyses reported here. Essentially, we required rats to leave the odor port area before presentation of the next sequence to help stereotype their approach behavior. The second reward port was used to train and maintain that requirement. We clarified this in the methods (P23,L15).

Supplementary

General: my previous remarks about stating the number of samples for each analysis, the nature of samples that the analyses are computed on, what do the error bar means etc. also apply to supplementary figures.

This information was added to the corresponding captions and methods, as well as in supplementary tables.

SF1: The three first diagonal plots look exactly identical, with the exception of some stretched / duplicated rows. This is very unexpected for biological data collected in different conditions, in this case different inter-trials. How do the authors explain this similarity?

We sincerely thank the reviewer for noticing this. The same plot was incorrectly duplicated in the first three diagonal panels when the figure was put together manually. We have corrected the figure.

SF2: How is the mean PSTH correlation computed? By PSTH do the authors mean matrix of PSTH? Are correlations computed between such matrices made from single trials? Please clarify or refer to the specific method section. The title “stimulus-specificity of sequential firing fields” could be more precise, like “evidence of time by odor coding”

The comment about the PSTH correlation approach has been addressed above (see P4,L21). We kept the original title to match the terminology used in the results and in Fig. 2.

SF5: what is T in the formula?

T refers to the matrix transposition operation (added to caption).

[SF5:] “The model was trained to minimize the difference between the original input data and the reconstructed data.” => is the model actually given the reconstructed data? In that case aren't the categories it finds constrained by what is chosen to be reconstructed? Perhaps the authors could explain more clearly what could be the pitfalls of this analysis and what controls they use to avoid those.

The reconstructed data (the output layer) is essentially a close copy of the original data (the input layer) -- it is *not* the data shown in Figure 3A. Briefly, the model is only given the original data (the input layer). The objective of the model is to iteratively adjust the weights of the layers to find the function that best reconstructs the original data into the output layer. So, by definition, the input and output layers will be very close -- they have to be within a specific reconstruction error threshold. The important thing here is that this process allows the bottleneck layer (the two nodes in the middle) to capture the essence of the data in two dimensions, which *is* what is shown in Figure 3A.

SF6: “used for the neural decoding analyses” -> Bayesian decoding was also used for some of the neural decoding analyses (time cells in Fig2) so this is a bit misleading

Although the term “decoding” could apply to any supervised model, in the paper we use the term “neural decoding” exclusively for the CNN model (Figs. 4, S4, S6). For the Bayesian model in Fig. 2, we use the terms “temporal coding” and “reconstructed time analyses”.

SF7: what does the “model training” overlay mean?

It is the time window from which neural activity was used to train the model. This information was added to the caption.

Review by Eleonore Duvelle.

References used:

Allen, Timothy A., Daniel M. Salz, Sam McKenzie, and Norbert J. Fortin. 2016. ‘Nonspatial Sequence Coding in CA1 Neurons’. *Journal of Neuroscience* 36 (5): 1547–63. <https://doi.org/10.1523/JNEUROSCI.2874-15.2016>.

- Carey, Alyssa A., Youki Tanaka, and Matthijs A. A. van der Meer. 2019. 'Reward Revaluation Biases Hippocampal Replay Content Away from the Preferred Outcome'. *Nature Neuroscience* 22 (9): 1450–59. <https://doi.org/10.1038/s41593-019-0464-6>.
- Davidson, Thomas J., Fabian Kloosterman, and Matthew A. Wilson. 2009. 'Hippocampal Replay of Extended Experience'. *Neuron* 63 (4): 497–507. <https://doi.org/10.1016/j.neuron.2009.07.027>.
- Kay, Kenneth, Jason E. Chung, Marielena Sosa, Jonathan S. Schor, Mattias P. Karlsson, Margaret C. Larkin, Daniel F. Liu, and Loren M. Frank. 2020. 'Constant Sub-Second Cycling between Representations of Possible Futures in the Hippocampus'. *Cell* 180 (3): 552-567.e25. <https://doi.org/10.1016/j.cell.2020.01.014>.
- Kay, Kenneth, Marielena Sosa, Jason E. Chung, Mattias P. Karlsson, Margaret C. Larkin, and Loren M. Frank. 2016. 'A Hippocampal Network for Spatial Coding during Immobility and Sleep'. *Nature* 531 (7593): 185–90. <https://doi.org/10.1038/nature17144>.
- Muessig, Laurenz, Michal Lasek, Isabella Varsavsky, Francesca Cacucci, and Thomas Joseph Wills. 2019. 'Coordinated Emergence of Hippocampal Replay and Theta Sequences during Post-Natal Development'. *Current Biology* 29 (5): 834-840.e4. <https://doi.org/10.1016/j.cub.2019.01.005>.
- Terada, Satoshi, Yoshio Sakurai, Hiroyuki Nakahara, and Shigeyoshi Fujisawa. 2017. 'Temporal and Rate Coding for Discrete Event Sequences in the Hippocampus'. *Neuron* 94 (6): 248-1262.e4. <https://doi.org/10.1016/j.neuron.2017.05.024>.
- Zheng, Chenguang, Ernie Hwaun, and Laura Lee Colgin. 2020. 'Impairments in Hippocampal Place Cell Sequences during Errors in Spatial Memory'. *BioRxiv*, April, 2020.04.20.051755. <https://doi.org/10.1101/2020.04.20.051755>.

Reviewer #3 (Remarks to the Author):

This paper by Shahbaba et al. presents exciting findings about fine-scale temporal dynamics and hippocampal sequence coding beyond the spatial domain. It is especially interesting to see predictive information within the ensemble code, and refreshing that these data are obtained during a memory task with clear indicators of trial by trial performance. The authors are employing appropriately complicated analysis – and do a good job of explaining the details of the analysis and stepping through interpretations of the data. I have a few specific questions about the data that I hope may add clarity and sometimes augment the ‘richness’ of the manuscript – but I have no major criticisms of the main findings of the study.

We thank the reviewer for the positive comments highlighting the significance of this challenging work, as well as the constructive suggestions that have allowed us to further improve the richness of the manuscript.

1. When looking at the peri-stimulus histograms in Figure 2a, one thing that jumps out is that during the first 250 ms, neural activity does not seem odor specific. E.g. – for the top row, plots 2-4 (which are ordered by sequential activation aligned to A) for the neurons that fired in the first 250 ms when presented A, those same neurons fire at high rates during the first 250 ms of presentation of B,C,D – of course the sequence is perturbed. This seems to be true for all four rows. Is there a subset of cells that are simply indicating stimulus onset? This seems to keep coming up – i.e. in Figure 3a, we see this beautiful sphere for stimulus information. I wonder if the authors might consider commenting on this early on (if my interpretation is true).

Yes, we noticed this as well. A subset of neurons increased their firing rate in the first time window (0:250ms) for all odors. Their firing is not precisely time-locked to port entry, suggesting they reflect a shared experience across trials (e.g., sampling the air in the port until the odor is identified). We had mentioned this when discussing the Figure 3 results and now added a mention of it earlier as suggested (with the Fig. 2 results; P5,L25).

2. Along these same lines, I am having a hard time reconciling these plots (in 2A) w/ the plot in 2D which shows the extended sequence across all odors. In this plot, I no longer see this propensity for some neurons to always fire within the first 250 ms of odor presentation. Is this because this is showing the ‘median’ onsets?

Yes, it is because we are showing median onsets (the task is self-paced so the interval between odors varies a bit). Re-aligning the data in Fig. 2D to the *actual* onset of odors BCDE would essentially reproduce Fig. 2A, albeit with a bit less power (because Fig. 2D includes fewer trials as it is focusing on full sequences of InSeq items).

3. The analysis of data presented in Figure 3 are compelling. It’s cool to see the cluster separation dynamics for different trial parameters – and especially neat to see some separation emerging early (like for Temporal Order and Trial Outcome). In the text, the authors point out that classification accuracy for Trial outcome and Temporal order are already significant between 250-500 ms. Would these be significant if a 99% cutoff were used? How do the authors interpret that Trial outcome classification falls again in the next temporal window despite improved Temporal order accuracy? Have the authors explored whether there are additional parameters represented in the hippocampal ensemble such as odor presentation and reward/no reward? I wonder if the increase seen in the Trial outcome in the last temporal window comes from conflation with reward signaling.

Is the Temporal Order and Trial Outcome differentiation in the 250-500ms still significant if 99% confidence intervals are used? The Trial Outcome effect still is, but the Temporal Order effect just falls short. This has been added in the text (P30,L6), but we kept the 95% CI results as the main focus as it is the most widely used confidence level.

Why does Trial Outcome go down in the 500-750ms window? We believe it is related to the fact that the original Fig. 3 did not match behavior across InSeq and OutSeq trials (it showed all consecutive 250ms windows to highlight the dynamics). Therefore, the 500:750ms window captured heterogeneous states across conditions (actual or impending withdrawals on correct OutSeq trials, maintenance of hold response on correct InSeq trials, and the reverse on incorrect trials), leading to increased variance in the data and reduced classification accuracy. The updated Fig. 3 (in response to Rev1,#1) is better because it allows us to show the dynamics while controlling for differences in behavior (i.e., it only shows time windows in which behavior is matched across trials: 0:250ms and 250:500ms relative to port entry, -250:0ms and 0:250ms relative to port withdrawal). The pattern of Trial Outcome differentiation is more consistent in the new figure.

Trial Outcome vs feedback signals. We expanded on the relationship between Trial outcome differentiation and the reward signaling in the results (P8,L10-17). Briefly, we view outcome differentiation during odor presentations as capturing the presence of a pattern in the ensemble activity that is predictive of whether the animal will respond correctly or not on that particular trial, which may reflect the anticipation of the associated reward/error signal, or disrupted representations of the predicted stimulus, currently presented stimulus, or InSeq/OutSeq status of the trial.

However, the fact that this differentiation peaks after the withdrawal response suggests that it then also incorporates the neural response to feedback signals (water reward on correct trials, buzzer on incorrect trials).

Examining the dynamics of additional parameters would require a different class of models. We plan to develop models to quantify latent state changes within trials to more closely examine the dynamics of each type of information (to see if we can further differentiate odor presentation from odor identity, and Trial Outcome from Feedback signals), but it is a considerable statistical challenge and the models will not be ready for quite some time. Therefore, for the present paper, we focused on these three key dimensions (defined *a priori*) using 250ms windows. Because such deep learning models are data-hungry, we are at the limit in terms of temporal resolution -- we tried smaller windows but the patterns and dynamics were not clearer.

4. When seeing the data about temporal compression of odor sequence, I immediately wondered if neurons are showing theta phase precession during this task. Are they?

Yes! Although the complexity of our design is not well-suited for examining phase precession (our use of many odors makes the odor-specific coding more graded and limits the number of trials on each), we did observe a proportion of neurons with significant phase precession (26.1% of neurons when collapsing data across BCD trials; 17.7%, 14.1%, and 16.8% when examining B, C, and D trials separately; see examples in new Fig. S5). This information was added to the results (P13,L17-23) and methods (P34,L28).

To examine this further, we also adapted the single-cell visualization method described below (see #5), in which the model was run on each neuron independently, and showed that the information coded by individual neurons significantly precessed across cycles (i.e., peak decoding of the present stimulus shifts from a late to an earlier phase within a trial). This new finding is now shown in Fig. 5H and described in the results (P13,L10-17) with the corresponding methods added to P35,L1.

5. Is it possible to visualize an example of the theta sequences with raw data? I.e. replace Figure 5a w/ trial data showing a theta cycle w/ rasterized spike activity of neurons that had peak activation for the different odors?

Unlike place cell activity, odor selectivity is more of a gradient than an all-or-none phenomenon. While we do see the past-present-future sequential pattern using rasterized activity from subsets of odor-selective cells (as suggested), we are concerned that this visualization might be mischaracterizing the type of patterns actually detected by the model (which are more likely to be subtle but distributed across many cells). We thought about this a lot and realized the best way to visualize the “raw data” at the individual neuron level is by focusing on the information coded by each neuron at each moment across the theta cycle (see new panel Fig. 5B and corresponding information in results and methods; P11,L21-30 and P34,L9, respectively). We feel this approach better captures what the model is actually detecting. Thank you for bringing this up -- although it was difficult to find the right solution, this visualization is a great addition to the paper.

6. Tying the theta sequence coding to the previous results – within a trial, the distinction of InSeq/OutSeq increased substantially between 500ms and 750ms – were theta sequences more robust during that time? (It is stated in the methods that this analysis was done on the first theta cycle beginning 100 ms after odor onset – so I guess I’m wondering if [they] could explore theta sequences throughout the trial – which could potentially link theta sequences to decisions about whether the odors are in the proper sequence). Were theta sequences present during OutSeq trials? Were there differences in theta sequences between correct and incorrect trials?

We considerably expanded Figure 5 and the corresponding results (P11-13) and methods (P33-35) sections to answer these important questions. More specifically, we used the same model training as before (trough of first theta cycle 100ms after port entry; correct InSeq trials only) but tested the model across other trial types and cycles. Note that, because of the need to match behavior across conditions, we had to carefully identify the proper window and subset of trials to target each question (see below).

Link with decision accuracy (correct vs incorrect trials). As with the other models, we examined the effect of accuracy using OutSeq trials (for better sampling of correct and incorrect trials) using the 250ms time window preceding port entry (to match behavior and avoid the potential influence of the OutSeq stimulus itself). Consistent with the prediction that the ascending phase is linked with the processing of upcoming information, we found that decoding of the expected (InSeq) stimulus in that phase was significantly higher on correct than incorrect trials (see new Fig. 5C). This information was added to the results (P12,L5) and methods (P33,L23).

InSeq vs OutSeq. We view this contrast as a way to examine the flexibility of this form of theta sequence information. Note that we could not test this comparison in the suggested window (500-750ms) because of differences in behavior (animals would still be in the port on InSeq trials, but may have correctly withdrawn on OutSeq trials). Rather, we compared the decoding of past, present, and future items across the three phases of the first cycle (cycle 1; see new

Fig. 5D). If the same pattern were observed between InSeq and OutSeq trials, it would indicate that these theta sequences rigidly reflect the most common sequence of items (InSeq) even if a different (OutSeq) stimulus is actually presented; instead, we observed the decoding was significantly different but only in the trough. The significantly lower decoding in the trough indicates the activity on OutSeq trials did not simply represent the same information as on InSeq trials (i.e., it reflected the OutSeq nature of the trial), whereas the lack of a significant difference on the descending and ascending phases suggests the same past and future InSeq stimuli were coded on InSeq and OutSeq trials which is consistent with task contingencies (i.e., items preceding and following OutSeq items are always InSeq). This information was added to the results (P12,L12) and methods (P33,L33).

Other theta cycles throughout the trial. Again, we used the same model training period as before but now tested the model across a series of four equidistant cycles, ranging from a pre-stimulus cycle to a cycle near the end of the odor presentation (cycle 5). Notably, we z-normalized decoding values of past, present, and future stimuli separately to highlight their respective magnitude and dynamics across phases and cycles (though the same pattern of results was observed without normalization). When focusing on the overall pattern on each cycle (collapsing across phase), we observed a structure consistent with the sequential reactivation reported with the CNN model (Fig. 5F). More specifically, we found that decoding of past, present and future stimuli exhibited significantly different distributions across cycles: past stimulus decoding increased toward the pre-stimulus cycle, present stimulus decoding peaked in cycle 1, and future stimulus decoding increased toward the last cycle. As mentioned above (#4), we also found that the information coded by individual neurons significantly precessed across cycles (Fig. 5H). This information was added to the results (P12,L28) and methods (P33,L10).

Taken together, these key new results significantly advance our understanding of the new form of theta sequence information we report by demonstrating it is important to perform accurate order judgments and that it flexibly captures trial-specific information associated with task demands. Critically, the pattern observed across cycles parallels the sequential reactivation phenomenon reported with the CNN model (compare Fig. 5F with Fig. 4D), demonstrating key areas of convergence between the models.

Minor: Is odor onset and port entry the same? If so, please only use one in the main text.

They are practically the same: port entry triggers the odor delivery so they are separated by 13 ms on average. We now only refer to “port entry” because it is the actual timestamp used to align the analyses. The change has been made to the text, figures, and figure captions. Thank you for catching that.

Minor: Consider citing Arinov, ... Tank 2017, to reference other non-spatial sequence coding in hippocampus

Yes, thank you. That citation was cut when we shortened the paper, but we found a good spot to add it back.

REVIEWERS' COMMENTS:

Reviewer #1 (Remarks to the Author):

I regret to hear about the medical condition described, and hope things are better now.

I went through the updated manuscript and rebuttal, and I am happy with the changes.

I have no further concerns.

Reviewer #2 (Remarks to the Author):

NCOMMS-20-21908A

Hippocampal ensembles represent sequential relationships among an extended sequence of nonspatial events

Shahbaba, Li, Agostinelli, Saraf, Cooper, Haghverdian, Elias, Baldi, and Fortin

This revised manuscript is greatly improved with new analyses, controls, clarifications, visualizations, and highlights of the new contributions, which together make the main message even more convincing, novel, and interesting. I would like to congratulate the authors for this important contribution at the interface of the memory and decision-making fields and only have a few (minor and optional) suggestions:

1. As previously mentioned, I think the manuscript would be more accurate if the new findings were not described as 'replay', which is strongly associated with the sequential reactivations that often occur during sharp-wave / ripples, when an animal is resting or pausing, i.e. disengaged from the external environment; instead, the findings are closest to 'theta sequences', which occur while the local field potential is dominated by theta rhythm and while the animal is running or seemingly engaged in a decision-making process (see how these terms are used e.g. in recent reviews: Pfeiffer, 2020; Drieu and Zugaro, 2019; Findlay et al., 2020). While I certainly agree that some of the findings are not equivalent to theta sequences (e.g. Fig 4), they are not similar to replay either, which takes place on a very fast timescale of the order of 10s to a few 100s of milliseconds. The authors mentioned that they used the term 'theta-associated replay' employed in Wickenheiser & Redish (2014) but it is unclear what study they refer to. What this wording may be reminiscent of instead is the phenomenon described by O'Neill

et al., 2006 (<https://www.sciencedirect.com/science/article/pii/S089662730500961X>) which shows co-firing during 'exploratory' sharp-wave / ripples, sometimes even in conjunction with a theta oscillation. However, this is not what the authors are interested in here and the analyses and changes implemented in this revised version have made this clearer, namely, showing the surprisingly low occurrence of sharp-wave / ripples and replacing 'replay' by 'reactivations'. For these reasons I feel that the choice to retain the wording 'replay' when used in conjunction with 'theta-associated' may confuse the message and would prefer if all mentions of 'replay' were replaced by 'reactivations' or another, less loaded, term. Of course, if the authors feel strongly about using 'replay' they could justify this view in a few sentences in the discussion.

2. The fact that most of the analyses were made while the rats were nose-poking, meaning that their behaviour would have to be almost identical at these times, is very important and a nice consequence of the study design. For extra clarification, the authors could explicitly mention (e.g. around p4, l10-14) that the reason why they state that behaviour was consistent across comparisons is because rats were nose-poking for the entire duration of odour presentations for most analyses.

3. Thank you for clarifying that only 1 session was used per animal. Could the authors explain how this single session was selected in methods (maybe around p25)?

4. The Zheng et al. study which was previously a preprint is now published in a peer-reviewed journal and its reference could be updated: Zheng, C., Hwaun, E. & Colgin, L. 2021, <https://www.nature.com/articles/s41467-021-23765-x>

5. P6 L27, "Because nonspatial stimuli are less strongly represented in hippocampal neurons than spatial location": to support this claim, the authors could add a reference to the recent review of Krupic & O'Keefe, 2021 (<https://journals.physiology.org/doi/full/10.1152/physrev.00014.2020>)

6. The reference Tang et al 2021 <https://elifesciences.org/articles/66227> could be added in discussion with the other references related to the interaction between hippocampus and prefrontal cortex during sequential decision-making (p16 l12)

7. Fig 5: the schematic with 'past stim. – present stimulus – future stim.- seems a bit out of place, and overlaps with the dashed lines radiating from 'cycle 1'; maybe moving this timeline at the top of the figure would help?

8. p25 l13 "assuming a minimum refractory period of 1 ms": are the authors sure this is what they used? The minimal refractory period value is usually 2ms for hippocampal rodent neurons.

Reviewer #4 (Remarks to the Author):

The authors have addressed all my comments.